



# Seasonal updraft speeds change cloud droplet number concentrations in low level clouds over the Western North Atlantic

Simon Kirschler[1,2], Christiane Voigt[1,2], Bruce Anderson[3], Ramon Campos Braga[4], Gao Chen[3], Andrea F. Corral[5], Ewan Crosbie[3], Hossein Dadashazar[5], Richard A. Ferrare[3], Valerian Hahn[1,2], Johannes Hendricks[1], Stefan Kaufmann[1,2], Richard Moore[3], Mira L. Pöhlker[6,7], Claire Robinson[3], Amy J. Scarino[3], Dominik Schollmayer[1,2], Michael A. Shook[3], K. Lee Thornhill[3], Edward Winstead[3], Luke D. Ziemba[3], and Armin Sorooshian[5,8]

[1]Institut für Physik der Atmosphäre, Deutsches Zentrum für Luft- und Raumfahrt (DLR), Oberpfaffenhofen, Germany
[2]Institut für Physik der Atmosphäre, Johannes Gutenberg-Universität, Mainz, Germany
[3]NASA Langley Research Center, Hampton, VA, USA
[4]Multiphase Chemistry Department, Max Planck Institute for Chemistry, Mainz, Germany
[5]Department of Chemical and Environmental Engineering, University of Arizona, Tucson, Arizona, USA
[6]Experimental Aerosol and Cloud Microphysics Department, Leibniz Institute for Tropospheric Research, Leipzig, Germany
[7]Faculty of Physics and Earth Sciences, Leipzig Institute for Meteorology, University of Leipzig, Leipzig, Germany
[8]Department of Hydrology and Atmospheric Sciences, University of Arizona, Tucson, Arizona, USA

**Correspondence:** Kirschler Simon (Simon.Kirschler@dlr.de)

**Abstract.** Low level clouds over the Western North Atlantic show a seasonal cycle in cloud properties which anticorrelates to aerosol concentrations. To determinate the impact of dynamic and aerosol processes within marine low clouds we examine the seasonal impact of updraft speed $w$ and cloud condensation nuclei concentration at 0.43% supersaturation ($N_{CCN_{0.43\%}}$) on the cloud droplet number concentration ($N_C$) of low level clouds over the Western North Atlantic Ocean. Aerosol and cloud

5   properties were measured with instruments on board the NASA LaRC Falcon HU-25 during the ACTIVATE (Aerosol Cloud meTeorology Interactions oVer the western ATlantic Experiment) mission in summer (August) and winter (February-March) 2020. The data are grouped in different $N_{CCN_{0.43\%}}$ loadings and the density functions of $N_C$ and $w$ near the cloud bases are compared. For low updrafts ($w < 1.3\,m\,s^{-1}$), $N_C$ in winter are mainly limited by the updraft speed and in summer additionally by aerosols. At larger updrafts ($w > 3\,m\,s^{-1}$), $N_C$ are impacted by the aerosol population, while at clean marine conditions

10   cloud nucleation is aerosol limited and for high pollution it is influenced by aerosols and updraft. The aerosol size distribution in winter shows a bimodal distribution in clean marine environments, which transforms to a unimodal distribution in high pollution levels due to altering processes, whereas unimodal distributions prevail in summer with a significant difference in their aerosol concentration and composition. The increase in pollution level is accompanied with an increase of organic aerosol and sulfate compounds in both seasons. We demonstrate that $N_C$ can be explained by cloud condensation nuclei activation

15   through upwards processed air masses with varying fractions of activated aerosols. The activation highly depends on $w$ and thus supersaturation between the different seasons, while the aerosol size distribution additionally affects $N_C$ within a season. Our results quantify the seasonal influence of $w$ and $N_{CCN_{0.43\%}}$ on $N_C$ and can be used to improve the representation of low marine clouds in models.



## 1 Introduction

Understanding cloud formation processes and their influence on the Earth's climate system are fundamental to assess climate model forecast quality (Zelinka et al., 2014, 2017; Seinfeld et al., 2016; IPCC, 2021). The results of the model evaluation activities of the Coupled Model Intercomparison Project Phase 6 (CMIP6) show that improvements in cloud representation result in stronger shortwave cloud feedbacks and higher effective climate sensitivity to the global mean surface air temperature of the CMIP6 model ensemble (Bock et al., 2020). In particular, regions with large multi model mean biases in near-surface air temperature and their cloud feedback are of high interest (Andrews et al., 2015; Ceppi et al., 2017) and targeted by various field campaigns (e.g., Lu et al., 2007; Hersey et al., 2009; Wood et al., 2011; Russell et al., 2013; Knippertz et al., 2015; Wendisch et al., 2016; Flamant et al., 2018; Sorooshian et al., 2018; Formenti et al., 2019; Sorooshian et al., 2019).

Atmospheric aerosols can act as cloud condensation nuclei (CCN) and activate to cloud droplets in favourable conditions, determined by atmospheric ambient parameters such as supersaturation and aerosol size and chemical composition (Köhler, 1936; Twomey, 1959; Koehler et al., 2006; Reutter et al., 2009; Rosenfeld et al., 2014; Cecchini et al., 2017; Prabhakaran et al., 2020). This leads to an alteration of cloud droplet number concentration $N_C$ (Twomey and Warner, 1967) and consequential cloud radiative effects (Twomey, 1977; Rosenfeld et al., 2019). Higher aerosol concentrations and additional CCN activation increases cloud lifetime and thickness by suppressing precipitation (Albrecht, 1989; Freud and Rosenfeld, 2012; Braga et al., 2017b). There are several approaches to quantify $N_C$ with satellite measurements. A direct approach utilizes the cloud optical depth, the cloud droplet effective radius and cloud top temperature (Grosvenor et al., 2018). An indirect approach exploits the aerosol-cloud interaction and uses the aerosol optical depth (AOD) as a proxy for $N_C$ (Quaas et al., 2008). (A list of symbols and abbreviations is given in Appendix A1.) Both approaches have high uncertainty, i.e. retrieving $N_C$ with AOD from satellites remains a challenge (Gryspeerdt et al., 2017; Painemal et al., 2020). Rosenfeld et al. (2016) have shown that based on a satellite methodology it is possible to retrieve cloud base $N_C$ and supersaturation, which further yields the CCN concentration at a given supersaturation with an accuracy of $\pm30\%$. However, satellites measure bulk properties which are limited in observing mechanisms on a microphysical scale (McComiskey et al., 2012). Consequently, in-situ measurements are needed to validate and enhance the understanding of the respective cloud processes.

This work focuses on the Western North Atlantic Ocean (WNAO) (Sorooshian et al., 2020) which provides ideal conditions for studying aerosol cloud interactions due to influence from the polluted East Coast of North America. Dadashazar et al. (2021b) find an anti-correlation in the seasonal cycle of AOD and $N_C$ for this area, which is in contrast to findings in other regions (e.g., Penner et al., 2006, 2011; Quaas et al., 2008; Gryspeerdt et al., 2016). Braga et al. (2017a) use a statistical approach (Haddad and Rosenfeld, 1997) to quantify the relationship of $w$ to $N_C$ at cloud bases of convective clouds over the Amazon basin. Braga et al. (2021) show good agreement of the derived relationship with an adiabatic parcel model. Our analysis focused on $w$ and aerosol impact on $N_C$ in marine boundary layer stratus and stratocumulus clouds near cloud base.



Global aerosol-climate simulations still suffer from large uncertainties in the representation of aerosol-cloud-radiation in­teractions (e.g., Myhre et al., 2013). Particularly large model uncertainties persist with regard to aerosol effects on marine clouds (e.g, McCoy et al., 2020, 2021). Simulating aerosol-cloud interactions in such models requires the application of micro­physical two-moment cloud schemes in combination with aerosol sub-models providing information about aerosol properties relevant for cloud formation (e.g., Lohmann et al., 2007; Lohmann and Hoose, 2009; Righi et al., 2020). Aerosol effects on $N_C$ are described in these models on the base of dedicated parameterizations (e.g., Abdul-Razzak and Ghan, 2000; Ghan et al., 2011) which are driven by model information about the aerosol size distribution and composition as well as $w$. Comparisons with observational data are essential to evaluate the robustness of simulating these quantities as well as the resulting $N_C$. The present study provides consistent information about all of these quantities under marine conditions for different seasons. Hence it is a valuable contribution to the data base available for global aerosol-climate model evaluation and can, therefore, trigger important improvements of aerosol-climate simulations and the applied parameterizations of the cloud nucleation process.

In the following sections we show that the aerosol size distribution in combination with $w$ determine $N_C$ near cloud base of marine clouds regardless of thermodynamic conditions. Furthermore, we found that the aerosol size distribution indicates the availability of CCN from the aerosol population and $w$ with the corresponding supersaturation signifying the fraction of activated CCN over the WNAO.

## 2 Methods

### 2.1 Region of study during the ACTIVATE campaign

The Aerosol Cloud meTerology Interactions oVer the western ATlantic Experiment (ACTIVATE) campaign (Sorooshian et al., 2019) is focused on probing clouds between $25° - 50°$N and $60° - 85°$W. Clouds are characterized simultaneously by the low flying NASA Langley Falcon HU-25 and the King Air UC-12 flying above. The Falcon HU-25 provides detailed in-situ measurements of aerosol, cloud, gas and meteorological properties by sampling the marine boundary layer (MBL) at different altitudes down to $150\,m$ above sea level, while the UC-12 probed clouds with remote sensing instruments flying at $8 - 10\,km$. The ACTIVATE mission plans for flights in three consecutive years (2020-2022) with 150 joint research flights (RF) and a total amount of 600 flight hours per aircraft, where 40 RF (35 joint, 5 alone with Falcon HU-25) with around 130 flight hours per plane were conducted in the first year 2020. Here we use measurements from the Falcon HU-25 RF of 2020 shown in Figure 1.

### 2.2 Instrumentation

#### 2.2.1 Cloud measurements: The Fast Cloud Droplet Probe

The Fast Cloud Droplet Probe (FCDP) (O'Connor et al., 2008; Knop et al., 2021) manufactured by Stratton Park Engineering Company Incorporated (SPEC Inc.) is a forward-scattering probe, which counts single particles in the diameter size range of $1.5 - 50\,\mu m$. In this analysis we use only particles with diameters larger than $3\mu m$. The FCDP uses a laser beam at



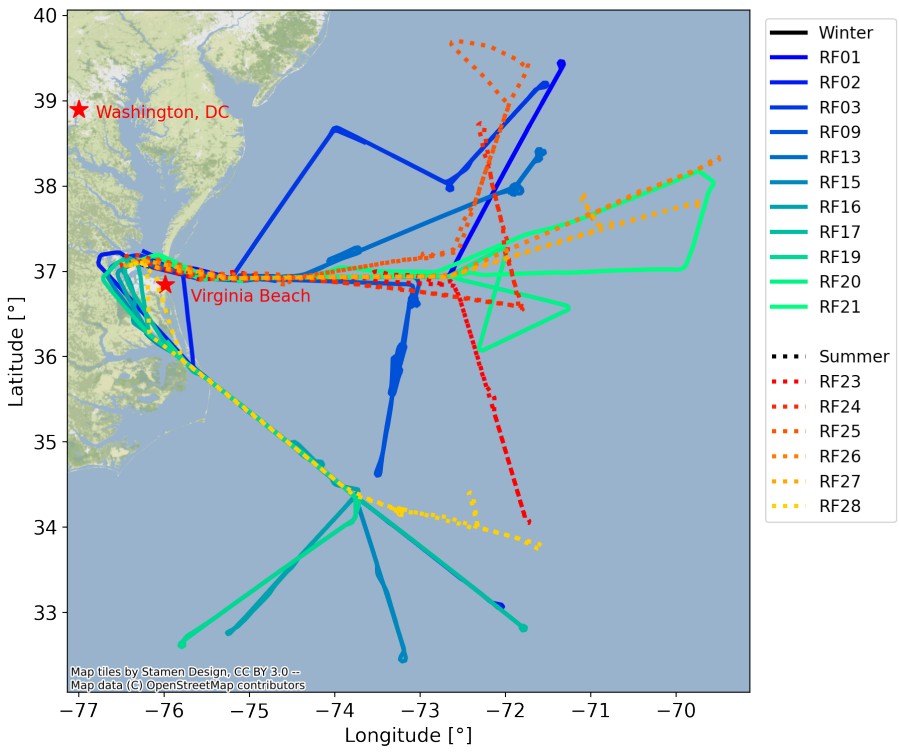

**Figure 1.** The used subset of HU-25 flights track during the first year of ACTIVATE flights in 2020. Each line represents a research flight with the running number of deployment. Lines colored with shades of blue to green represent flights during winter (February-March) while dotted orange lines indicate flights during summer (August).

785 $nm$ wavelength to collect light scattered by particles passing through the laser beam according to Mie theory in a $4° - 12°$ collection angle. A 70:30 beam splitter is used to split the collected light to a signal and qualifier detector. The signal detector has a 800 $\mu m$ pinhole for coincidence reduction (Lance, 2012) and a rectangular slit aperture with 800 $\mu m$ length and 200 $\mu m$ 85 width. Both detectors convert the incoming light intensity into corresponding voltages and amplify them over two stages. The beam diameter on the detectors depends on the distance of the measured particle from the focal plane of the collecting lens system. The ratio of the qualifier voltage to signal voltage is the so-called depth of field (DoF) criteria which can be used to limit the sample area of the probe, because the slit aperture width restricts the intensity on the qualifier detector depending on the magnification of the beam diameter. In this analysis we use a DoF criteria >0.6, which is equivalent to a calibrated 90 sample area of $0.248 \ mm^2$ (Lance et al., 2010; Faber et al., 2018). With a sampling rate of 25 $ns$ the FCDP additionally stores the transit time, inter-arrival time and waveform of each particle. These parameters are used for data corrections, see Baumgardner et al. (1985); SPEC inc (2012). Coincidence correction is applied by deriving a theoretical particle transit time, determined by particle air speed (PAS) and particle diameter, under consideration of a top hat intensity along the laser beam cross section. Measured particles with transit times larger than 125% of the theoretical transit time are deemed coincident and



are thus discarded. A shattering correction is done by using the adaptive method and a waveform symmetry filter is applied, both methods are described in SPEC inc (2012).

According to Baumgardner et al. (2017), light scattering probes have a propagated uncertainty in size due to Mie ambiguity, collection angles, coincidence, nonsphericity and shattering of $10 - 50\%$ and a propagated uncertainty in $N_C$ due to sample area uncertainty, coincidence and shattering of $10 - 30\%$ (Kleine et al., 2018; Bräuer et al., 2021a, b). The FCDP with its fast
electronics, small pinhole feature for coincidence reduction and applicable filtering techniques can be classified among the lower end of both propagated uncertainties in size and $N_C$.

### 2.2.2 Cloud measurements: The Two-Dimensional Stereo Probe

The Two-Dimensional Stereo probe (2D-S) from SPEC Inc. is an optical array probe, which generates shadow images of particles with a linear array of 128 photodiodes (Lawson et al., 2006, 2019). It measures single particles in a size range of
$5.7 - 1465 \ \mu m$ with an effective pixel size of $11.4 \ \mu m$ for each photodiode channel. The 2D-S has two identical subsystems perpendicularly aligned with a combination of transmitting and receiving arm each. Both arms operate with a laser of $785 \ nm$ wavelength and traversing particles generate a diffraction pattern according to diffraction theory. Similar to the FCDP the 2D-S has an optical plane, determined by the focal points of the light collecting lens system. Each photodiode is triggered if light intensity falls below a threshold of 50%. All shadowed photodiodes are recorded at a fast succession, specified by the sampling
rate, while a particle passes the laser beam. A recorded ensemble of slices produces a two-dimensional image of the particle (Knollenberg, 1970).

The sample area of the 2D-S depends on the particle size. The diffraction pattern can be calculated analytically with angular spectrum theory or fresnel theory (Korolev et al., 1991) and depends on the particles distance to the optical plane. With increasing distance the diffracted light forms spots of destructive interference and the particle is magnified. These particles
are classified as out-of-focus. In the case of spheroidal liquid droplets the so called poisson spot forms at the center of the projected circle and a size correction can be applied by relating the radius of the poisson spot to that of the captured particle image (Korolev, 2007). The magnification results in a decrease of the shadowing on the photodiodes until it is below the 50% threshold and no photodiode is triggered. The size dependent optical depth of field was verified for optical array probes according to Korolev et al. (1998). The maximum optical depth of field equals half of the $63 \ mm$ distance of the 2D-S
transmitting arm to the receiving arm. The maximum is reached with a drop radius of $109 \ \mu m$ and the maximum sample area can be calculated by multiplying the arm distance with the array width. In this work the all-in method is used to determine the effective array width and thus the sample area. The all-in method rejects particles with occulted edge photodiodes and adjusts the sample area depending on size, because the possibility of large particles rejected is higher compared to small particles (Knollenberg, 1970).

The 2D-S measures with a constant sampling rate resulting in an artificially elongated/shortened particle image if the actual PAS deviates from the PAS for which the sampling rate was computed. The PAS was measured by a pitot tube attached to a Cloud Aerosol and Precipitation Spectrometer (Voigt et al., 2017, 2021), which was mounted on the opposite wing at the same



position as the 2D-S/FCDP Combination. With the PAS to sampling rate ratio the deformed images can be corrected (Weigel et al., 2016).

Sizing accuracy is affected by out of focus and shattered particles, the time response and discretization of the probe hardware and lies for imaging probes in a $10-100\%$ range according to Baumgardner et al. (2017). While the uncertainty of out of focus spheroids is reduced with Korolev's correction it remains for ice particles. The 2D-S has a relative fast response time of $41\ ns$ and can be classified on the lower end of the uncertainty range for spheroids and in the middle for ice particles (Lawson and Baker, 2006; Baker and Lawson, 2006; Gurganus and Lawson, 2018). The concentration is affected by the size dependent

optical depth of field and shattering (Lawson, 2011). With shattering removal and an adjusted sample area the 2D-S here is similarly representative for the lower range of 10%-100% propagated uncertainty in $N_C$.

Measured size distributions of FCDP and 2D-S overlap in the size range of $16-51.3\ \mu m$. We perform the overlap calculation for the size range between the lower FCDP bin edge at $27\ \mu m$ to the upper 2D-S bin edge of $39.9\ \mu m$. The particle distribution inside the overlap 2D-S bin is estimated with the next 2D-S bin by linear interpolation and attributed proportionally to the last

FCDP bin and a new 2D-S bin.

### 2.2.3 Vertical velocity

The winds on the HU-25 are measured by the NASA Langley TAMMS (turbulent air motion measurement system). The primary components include fast-response flow-angle and temperature sensors to determine the wind with respect to the aircraft along with an Applanix 650 inertial navigation system (Applanix Inc.) to provide the aircraft's position, speed and altitude.

The data is recorded at 200 Hz on a UEIPAC-300 real-time controller (United Electronics Industries) and then averaged down to 20 Hz for processing, analysis, and data archiving. The flow-angle system includes five, flush-mounted pressure-ports installed in a cruciform pattern in the aircraft radome to provide angle of attack (vertically-aligned ports) and side-slip (horizontally aligned ports) measurements. Corresponding fast-response high-precision pressure transducers are placed as close as possible to the pressure ports in order to minimize delays and errors. Pitch and yaw maneuvers, speed variations and reverse

headings are performed periodically during deployments to verify system operation and calibration and validate derived mean horizontal-wind vectors. Three dimensional winds are computed from the full air motion equations (Lenschow, 1986). The aircraft platform velocity components are computed internally by the Applanix by combining the GPS and inertial data via a Kalman filtering technique. Ambient air temperature measurements needed to determine true air speed are made with a Rosemount Model 102 non-deiced total air temperature sensor with a fast response platinum sensing element (E102E4AL).

This setup has been used extensively for other campaigns on the NASA P-3 aircraft. Additional details on the instrumentation, calibration, and intercomparison results of the TAMMS when used on the NASA P-3 can be found at Thornhill et al. (2003). All wind measurements including horizontal and vertical winds have a 5% uncertainty.

### 2.2.4 Aerosol Measurements: Cloud Condensation Nuclei

The CCN number concentrations were measured with a CCN-100 counter manufactured by Droplet Measurement Technolo-

gies, which is based on the concept of Roberts and Nenes (2005) and characterized by Lance et al. (2006). The CCN-100 was





operated in two modes during ACTIVATE. The first is a continuous flow mode where ambient air enters a column shaped humidified chamber with a constant supersaturation of 0.43%. Aerosols are activated depending on their size and chemical properties. The droplets are measured afterwards by an optical particle counter. The second is a scanning flow mode where the flow rate in the chamber is changed while a constant temperature gradient is maintained (Moore and Nenes, 2009). Here

an aerosol sample is exposed to a continuously changing supersaturation in the chamber and the concentration of activated aerosols $N_{CCN}$ is measured depending on supersaturation. One scan is typically done in a $10 - 60$ seconds time interval and in this analysis we use the mean of $N_{CCN}$ in a supersaturation range of $0.40-0.46\%$ to appromximate $N_{CCN_{0.43\%}}$. The uncertainty in percent supersaturation is $\pm0.04$ and in $N_{CCN_{0.43\%}}$ $\pm10\%$.

Since the instrument supersaturation is fixed in continuous flow mode and artificially generated in scanning flow mode we

have to estimate the supersaturation in cloud base. The maximum supersaturation $S_{max}$ is calculated according to Pinsky et al. (2012) with

$$S_{max} = Cw^{\frac{3}{4}}N_c^{-\frac{1}{2}} \tag{1}$$

where C is determined by cloud base temperature and pressure, and $w$ and $N_C$ are the updraft speed and cloud droplet number concentration, respectively, measured in cloud base.

**2.2.5 Aerosol Measurements: Chemical Composition**

Submicron non-refractory aerosol chemical composition was measured by a High Resolution Time-of-Flight Aerosol Mass Spectrometer (HR-ToF-AMS; Aerodyne Research Inc. DeCarlo et al., 2006; Hilario et al., 2021). Mass concentrations of sulfate, nitrate, chloride, ammonium, and organic matter were recorded at 1Hz and averaged to 30-s for all subsequent analyses. Measurements were made isokinetically using a forward-facing dual-diffuser aircraft inlet (model 1200, Brechtel Manufac-

turing Inc.) and were pressure-controlled at 500 torr. Mass concentrations were processed using default relative ionization efficiencies for each chemical component, with a collection efficiency of unity, and are reported at standard temperature and pressure (STP; $273.15\ K$ and $1013.25\ mb$).

The particle-into-liquid sampler (PILS) obtained water-soluble aerosol composition data. Sampled aerosol particles were grown into droplets that were collected via inertial impaction and transported to vials on a rotating carousel. The liquid content

of the vials was analyzed post-flight via ion chromatography for water-soluble ions (Sorooshian et al., 2006). This study reports on sea salt concentrations by attributing $Na^+$, $Cl^-$ and $Mg^{2+}$ exclusively to sea salt and adding $SO_4^{2-}$ (0.25), $K^+$ (0.036) and $Ca^{2+}$ (0.039) fractions based on their ratio to $Na^+$ in sea water (Bowen, 1979; Farren et al., 2019; Ma et al., 2021). The PILS sample particles up to roughly $4\ \mu m$ in diameter (McNaughton et al., 2007; Hilario et al., 2021).

**2.2.6 Aerosol Measurements: Size Distribution**

Aerosol size distributions were obtained from a combination of two particles sizers. A custom Scanning Mobility Particle Sizer (SMPS; Differential Mobility Analyzer model 3085 and Condensation Particle Counter model 3776, TSI, Inc., Moore





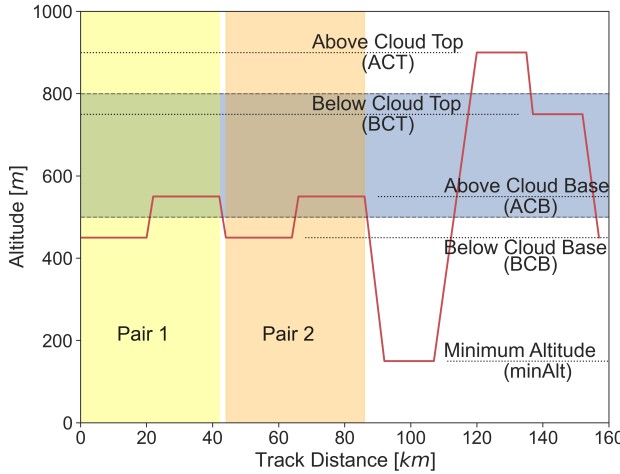

**Figure 2.** Typical flight pattern of one cloud ensemble of the HU-25 aerosol, cloud and vertical velocity data set. In this study we focus on the four legs in the beginning of each ensemble containing two pairs (yellow and orange shaded areas) consisting of a below cloud base leg followed by a above cloud base leg. The blue shading indicates the cloud layer.

et al., 2017) measured $3.2 - 89.1 \ nm$ diameter particle size distributions at approximately 60 s time response. A Laser Aerosol Spectrometer (LAS; model 3340, TSI, Inc.; Moore et al., 2021) measured $100 - 3162 \ nm$ diameter particles at 1Hz time response. SMPS sizing is calibrated and frequently verified using NIST-traceable polystyrene latex spheres. LAS sizing is calibrated using lab-generated monodisperse ammonium sulfate (refractive index = 1.52). Each instrument sampled dried air from the same common inlet as the HR-ToF-AMS and data are reported at STP.

### 2.3 Methodology

In this study we select the data a priori into pairs of series of below cloud base (BCB) and above cloud base (ACB) legs resulting in two pairs per ensemble (ensemble is a collection of legs below, in, and above clouds) flown during ACTIVATE, shown in Figure 2. This flight design intends for measurements to reflect the same environment. Closely spaced aerosol and cloud measurements are ensured by taking the latest full $N_{CCN}$ scan or 60 seconds of continuous $N_{CCN_{0.43\%}}$ measurements of the BCB leg and the last measurement of a cloud portion in the nearest ACB leg is restricted to never exceed a horizontal distance of $40 \ km$ to the aerosol measurement. Cloud periods are defined as seconds with a threshold of liquid water content $> 0.02 \ gm^{-3}$ and $N_C > 20 \ cm^{-3}$. We additionally excluded pairs with precipitation occurrences in the BCB leg by using the 2D-S size distribution and images, since the $N_{CCN_{0.43\%}}$ measurements are influenced by the large particles shattering on the aerosol inlet and precipitation indicates that the cloud is at a different point of its life cycle where agglomeration and coalescence altered $N_C$ and aerosol removal occurred below cloud. Each flight leg pair consists of a $N_{CCN_{0.43\%}}$ distribution taken from the pair's BCB leg either in continuous flow or scanning flow mode, a mean aerosol loading derived from the



$N_{CCN_{0.43\%}}$ distribution, $N_C$ and positive vertical velocity measurements (updraft speeds $w$) in cloud portions of the pair's ACB
leg.

For ensuring similar environmental conditions the pairs are classified with respect to their mean $N_{CCN_{0.43\%}}$ into a low pol-
luted (LP), medium polluted (MP) and high polluted (HP) group. For comparison both seasons share the boundaries separating
the groups and the bin boundaries are chosen by identifying modes in the distribution of all winter pair mean $N_{CCN_{0.43\%}}$ values.
The LP groups contains $N_{CCN_{0.43\%}}$ from the minimum measured to $372\ cm^{-3}$, the MP group extends from $> 372 - 769\ cm^{-3}$
and the HP group is defined for $> 769\ cm^{-3}$ to the maximum measured in the respective season. The Probability Matching
Method (PMM) is used on each group's set of $N_C$ and $w$ within a 2.5 to 97.5 percentile interval to quantify the impact of $w$ for
the different pollution levels.

We use the effective updraft speed $w_{eff}$ for approximating the updraft through the measured $w$ density function in cloud base

$$w_{eff} = \frac{\int w^2}{\int w}. \tag{2}$$

With the help of the $w$ to $N_C$ relation from the PMM the corresponding $N_C$ to $w_{eff}$ can be derived and therefore a $S_{max}$
estimate, which is representative for the supersaturation in cloud base of the respective group. We use the variability and
magnitude of $w$ with the related $S_{max}$ estimates, the aerosol size distribution and chemical composition to quantify their
contribution to the activation of CCN in the winter and summer season 2020 for different pollution levels.

### 2.3.1 Probability Matching Method

The PMM was proposed by Calheiros and Zawadzki (1987) for a statistical comparison of radar reflectivity to rain rate. The
derived relationship is verified and performs significantly better than power law regression (Rosenfeld et al., 1994). Addi-
tional improvements by taking physical parameters into account for different rain type classification were done by Rosenfeld
et al. (1995). The PMM is mathematically justified with an error estimation by Haddad and Rosenfeld (1997), and Braga et al.
(2017a) showed that the PMM can be applied to get a reasonable relationship of $w$ to $N_C$. The PMM is based on the assumption
that two related parameters taken in non-simultaneous measurements, sharing the same environment in terms of climatolog-
ical and physical means, are increasing monotonically with each other. The relationship can be computed by matching the
percentiles of the parameter's density functions, with more details on the mathematical background described in Haddad and
Rosenfeld (1997). Braga et al. (2021) showed good agreement between measurements of $N_C$ at cloud bases of convective
clouds and estimations from an adiabatic parcel model.

## 3 Results and Discussion

### 3.1 Measurements of $N_{CCN_{0.43\%}}$ below and $N_C$, and $w$ near Cloud Base

The measurements are grouped into pairs with consecutive BCB and ACB legs, illustrated in Figure 3a for a pair during RF02
on 15 February 2020. In Figure 3b the size distribution of the 2D-S/FCDP combination is shown. The FCDP measures a



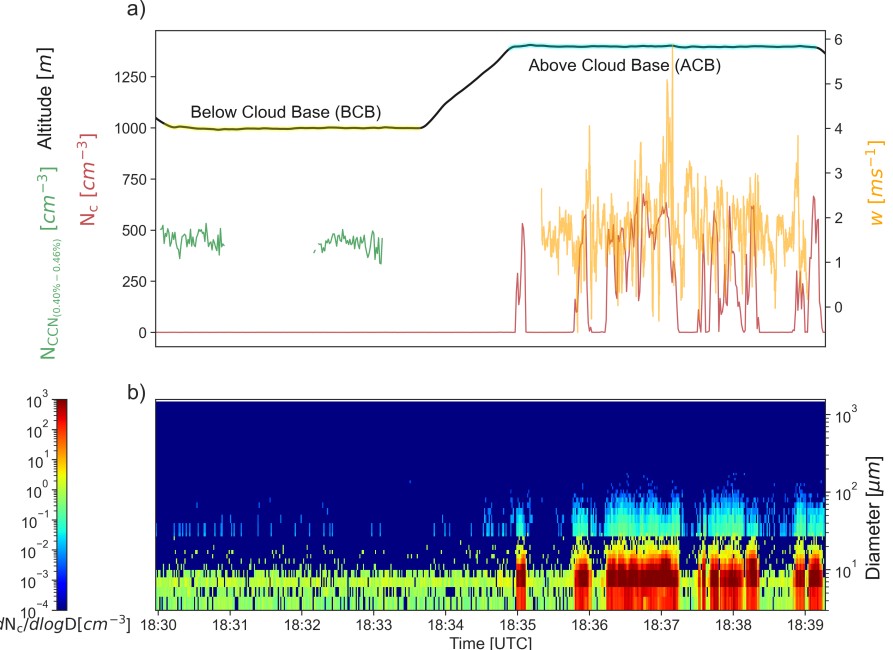

**Figure 3.** a) Aerosol and cloud properties measured in an ensemble pair of BCB and ACB legs from RF02 on 15 February 2020. The Falcon altitude is given by the black line and the yellow and blue shading indicates the BCB and the ACB legs. $N_C$ (red), $N_{CCN_{0.43\%}}$ (green) and $w$ (orange) are shown as lines. The aerosol loading $N_{CCN_{0.43\%}}$ representative of the pair's environment is $440(\pm 20)\ cm^{-3}$, calculated from the measurements mean within the BCB leg between $0.40 - 0.46\%$ supersaturation of the CCN-100 scans nearest to the cloud measurement. The pair's mean values with standard deviation for $w$ and $N_C$ are $1.85(\pm 0.82)\ ms^{-1}$ and $385(\pm 171)\ cm^{-3}$, respectively. b) Histogram showing the color-coded log-normalized number concentrations per bin on a 1-second basis of the 2D-S/FCDP combination with the diameter given in the ordinate.

constant particle background in a size range of $3 - 12\ \mu m$ with concentrations between 0.1 to 1 per cubic centimeter. The

background is visible in all BCB legs, but vanishes at flight levels above cloud top above the boundary layer (not shown). The relative humidity data show values mainly above 75% in the BCB legs for both seasons, suggesting the measured background concentrations being deliquescent sea salt particles.




**Table 1.** All pairs consisting of serial below cloud base (BCB) and above cloud base (ACB) legs during the February-March 2020 deployment. Mean values and standard deviation in parenthesis for $w$ and $N_C$ from ACB cloud portions, and $N_{CCN_{0.43\%}}$ from the BCB legs. $D_{max}$ is the maximal distance of cloud measurements to the aerosol measurements and $h_{ACB}$ is the height above cloud base with standard deviation in parenthesis.

| Flight | Date | $t_{initial}$ [UTC] | CCN-100 Mode (Supersat [%]) | $D_{max}$ [km] | in-cloud [s] | $N_{CCN_{0.43\%}}$ [$cm^{-3}$] | $w$ [$ms^{-1}$] | $N_C$ [$cm^{-3}$] | $h_{ACB}$ [m] |
|---|---|---|---|---|---|---|---|---|---|
| RF01 | 14 Feb 2020 | 17:21:32 | Scan (0.17-0.70) | 37.2 | 19 | 647 ($\pm$ 35) | 0.83 ($\pm$ 0.56) | 298 ($\pm$ 173) | 127 ($\pm$ 4) |
| RF01 | 14 Feb 2020 | 17:30:17 | Scan* (0.17-0.70) | 35.3 | 28 | 664 ($\pm$ 50) | 1.67 ($\pm$ 0.70) | 593 ($\pm$ 492) | 136 ($\pm$ 14) |
| RF01 | 14 Feb 2020 | 17:58:43 | Scan (0.16-0.69) | 28.3 | 51 | 582 ($\pm$ 46) | 1.74 ($\pm$ 1.21) | 723 ($\pm$ 344) | 103 ($\pm$ 5) |
| RF01 | 14 Feb 2020 | 18:05:17 | Scan* (0.16-0.68) | 35.6 | 44 | 582 ($\pm$ 36) | 2.07 ($\pm$ 1.26) | 570 ($\pm$ 308) | 111 ($\pm$ 3) |
| RF02 | 15 Feb 2020 | 17:09:31 | Scan (0.17-0.71) | 22.9 | 59 | 436 ($\pm$ 37) | 0.62 ($\pm$ 0.48) | 389 ($\pm$ 217) | 82 ($\pm$ 6) |
| RF02 | 15 Feb 2020 | 17:18:16 | Scan (0.17-0.71) | 19.8 | 58 | 630 ($\pm$ 36) | 0.63 ($\pm$ 0.33) | 648 ($\pm$ 279) | 73 ($\pm$ 2) |
| RF02 | 15 Feb 2020 | 18:23:53 | Scan (0.16-0.71) | 29.4 | 34 | 489 ($\pm$ 34) | 0.87 ($\pm$ 0.52) | 297 ($\pm$ 223) | 147 ($\pm$ 6) |
| RF02 | 15 Feb 2020 | 18:32:38 | Scan (0.16-0.71) | 38.4 | 130 | 440 ($\pm$ 20) | 1.85 ($\pm$ 0.82) | 385 ($\pm$ 171) | 200 ($\pm$ 3) |
| RF03 | 17 Feb 2020 | 17:41:11 | Scan* (0.17-0.71) | 40.0 | 74 | 1564 ($\pm$ 65) | 0.25 ($\pm$ 0.29) | 930 ($\pm$ 663) | 93 ($\pm$ 3) |
| RF09 | 27 Feb 2020 | 18:47:10 | Scan* (0.16-0.72) | 32.7 | 62 | 659 ($\pm$ 39) | 0.72 ($\pm$ 0.53) | 671 ($\pm$ 357) | 98 ($\pm$ 5) |
| RF09 | 27 Feb 2020 | 18:55:55 | Scan (0.17-0.72) | 29.7 | 36 | 575 ($\pm$ 46) | 0.64 ($\pm$ 0.53) | 336 ($\pm$ 218) | 125 ($\pm$ 6) |
| RF09 | 27 Feb 2020 | 19:28:43 | Scan (0.16-0.71) | 37.5 | 41 | 582 ($\pm$ 29) | 0.73 ($\pm$ 0.54) | 467 ($\pm$ 250) | 145 ($\pm$ 5) |
| RF09 | 27 Feb 2020 | 19:39:39 | Scan (0.17-0.71) | 33.2 | 48 | 656 ($\pm$ 42) | 0.91 ($\pm$ 0.77) | 355 ($\pm$ 224) | 189 ($\pm$ 19) |
| RF09 | 27 Feb 2020 | 20:10:17 | Scan (0.16-0.71) | 28.7 | 42 | 674 ($\pm$ 29) | 1.13 ($\pm$ 0.94) | 716 ($\pm$ 377) | 151 ($\pm$ 4) |
| RF09 | 27 Feb 2020 | 20:19:02 | Scan (0.16-0.71) | 31.9 | 35 | 650 ($\pm$ 35) | 0.83 ($\pm$ 0.68) | 647 ($\pm$ 292) | 199 ($\pm$ 4) |
| RF13 | 01 Mar 2020 | 14:10:32 | Scan* (0.16-0.71) | 28.2 | 96 | 1217 ($\pm$ 93) | 1.57 ($\pm$ 1.28) | 1020 ($\pm$ 556) | 113 ($\pm$ 4) |
| RF13 | 01 Mar 2020 | 15:00:51 | Scan (0.16-0.72) | 37.2 | 74 | 361 ($\pm$ 19) | 1.54 ($\pm$ 1.63) | 372 ($\pm$ 197) | 169 ($\pm$ 5) |
| RF13 | 01 Mar 2020 | 16:02:06 | Scan* (0.17-0.71) | 36.7 | 51 | 769 ($\pm$ 41) | 1.46 ($\pm$ 1.30) | 818 ($\pm$ 721) | 139 ($\pm$ 3) |
| RF16 | 06 Mar 2020 | 19:34:26 | Scan (0.17-0.71) | 32.3 | 55 | 991 ($\pm$ 46) | 0.99 ($\pm$ 0.73) | 1367 ($\pm$ 958) | 208 ($\pm$ 6) |
| RF16 | 06 Mar 2020 | 19:43:11 | Scan (0.16-0.72) | 28.3 | 36 | 1788 ($\pm$ 109) | 1.80 ($\pm$ 1.06) | 1157 ($\pm$ 912) | 100 ($\pm$ 3) |
| RF16 | 06 Mar 2020 | 20:15:59 | Scan (0.17-0.72) | 29.9 | 49 | 1501 ($\pm$ 71) | 1.84 ($\pm$ 1.06) | 1014 ($\pm$ 742) | 130 ($\pm$ 6) |
| RF16 | 06 Mar 2020 | 20:24:44 | Scan* (0.17-0.72) | 34.7 | 33 | 945 ($\pm$ 53) | 1.55 ($\pm$ 1.27) | 397 ($\pm$ 358) | 193 ($\pm$ 5) |
| RF17 | 08 Mar 2020 | 14:34:49 | Flow (0.43) | 25.0 | 39 | 183 ($\pm$ 28) | 1.05 ($\pm$ 0.88) | 434 ($\pm$ 228) | 117 ($\pm$ 3) |
| RF17 | 08 Mar 2020 | 14:44:29 | Flow (0.43) | 29.2 | 17 | 245 ($\pm$ 31) | 1.35 ($\pm$ 0.68) | 498 ($\pm$ 214) | 135 ($\pm$ 3) |
| RF17 | 08 Mar 2020 | 15:11:45 | Flow (0.43) | 28.7 | 112 | 164 ($\pm$ 26) | 0.46 ($\pm$ 0.45) | 208 ($\pm$ 93) | 173 ($\pm$ 4) |
| RF17 | 08 Mar 2020 | 15:23:22 | Flow (0.43) | 28.3 | 72 | 96 ($\pm$ 18) | 0.95 ($\pm$ 0.98) | 218 ($\pm$ 101) | 163 ($\pm$ 4) |
| RF17 | 08 Mar 2020 | 15:52:58 | Flow (0.43) | 30.1 | 56 | 196 ($\pm$ 27) | 0.83 ($\pm$ 0.85) | 386 ($\pm$ 212) | 91 ($\pm$ 3) |
| RF17 | 08 Mar 2020 | 16:02:17 | Flow (0.43) | 19.8 | 65 | 225 ($\pm$ 33) | 1.52 ($\pm$ 1.34) | 346 ($\pm$ 149) | 129 ($\pm$ 4) |
| RF19 | 09 Mar 2020 | 17:27:27 | Flow (0.43) | 28.6 | 26 | 291 ($\pm$ 34) | 0.63 ($\pm$ 0.48) | 208 ($\pm$ 146) | 125 ($\pm$ 6) |
| RF19 | 09 Mar 2020 | 17:57:47 | Flow (0.43) | 23.7 | 20 | 299 ($\pm$ 44) | 0.61 ($\pm$ 0.44) | 247 ($\pm$ 125) | 121 ($\pm$ 4) |





| Flight | Date | $t_{\mathrm{initial}}$ [UTC] | CCN-100 Mode (Supersat [%]) | $D_{\max}$ [$km$] | in-cloud [$s$] | $N_{\mathrm{CCN}_{0.43\%}}$ [$cm^{-3}$] | $w$ [$ms^{-1}$] | $N_{\mathrm{C}}$ [$cm^{-3}$] | $h_{\mathrm{ACB}}$ [$m$] |
|---|---|---|---|---|---|---|---|---|---|
| RF19 | 09 Mar 2020 | 18:41:47 | Flow (0.43) | 17.5 | 18 | 335 ($\pm$ 46) | 0.43 ($\pm$ 0.28) | 215 ($\pm$ 114) | 224 ($\pm$ 2) |
| RF19 | 09 Mar 2020 | 18:50:13 | Flow (0.43) | 37.4 | 24 | 307 ($\pm$ 36) | 0.64 ($\pm$ 0.56) | 285 ($\pm$ 171) | 196 ($\pm$ 11) |
| RF20 | 11 Mar 2020 | 13:46:55 | Flow (0.43) | 25.3 | 22 | 875 ($\pm$ 101) | 0.45 ($\pm$ 0.46) | 780 ($\pm$ 430) | 62 ($\pm$ 2) |
| RF20 | 11 Mar 2020 | 14:26:13 | Flow (0.43) | 23.3 | 10 | 986 ($\pm$ 134) | 0.26 ($\pm$ 0.21) | 320 ($\pm$ 221) | 42 ($\pm$ 3) |
| RF21 | 12 Mar 2020 | 14:43:10 | Flow (0.43) | 25.2 | 19 | 586 ($\pm$ 84) | 1.64 ($\pm$ 1.07) | 675 ($\pm$ 383) | 141 ($\pm$ 4) |
| RF21 | 12 Mar 2020 | 14:51:22 | Flow (0.43) | 27.4 | 30 | 500 ($\pm$ 91) | 0.77 ($\pm$ 0.70) | 458 ($\pm$ 275) | 140 ($\pm$ 4) |
| RF21 | 12 Mar 2020 | 15:19:57 | Flow (0.43) | 21.2 | 42 | 587 ($\pm$ 102) | 0.78 ($\pm$ 0.72) | 654 ($\pm$ 418) | 71 ($\pm$ 2) |
| RF21 | 12 Mar 2020 | 16:06:00 | Flow (0.43) | 22.3 | 34 | 494 ($\pm$ 58) | 0.68 ($\pm$ 0.48) | 559 ($\pm$ 255) | 116 ($\pm$ 3) |
| RF21 | 12 Mar 2020 | 16:14:01 | Flow (0.43) | 38.8 | 25 | 455 ($\pm$ 72) | 0.69 ($\pm$ 0.57) | 584 ($\pm$ 261) | 124 ($\pm$ 46) |
| All RF | Average | | | 29.3 | 45 | 612 | 1.02 | 535 | 130 |

\* Only one scan in BCB leg.

All data ensemble pairs used in this study from the ACTIVATE winter February-March 2020 deployment are given in Table 1. We use selected pairs with a minimum in-cloud time above or equal 10 seconds for sufficient statistics. $N_{\mathrm{C}}$ is predicted to reach its maximum at a height above cloud base depending on $w$ and subsequent $S_{\max}$ estimates in an adiabatic parcel model (Braga et al., 2021). That the activation of CCN into cloud droplets had sufficient time is ensured by taking only pairs into account with cloud measurements at a height above cloud base $h_{\mathrm{ACB}}$ greater than $35\ m$ in this analysis. The $h_{\mathrm{ACB}}$ is gauged by calculating the middle of the difference between leg-mean values at BCB and ACB altitudes. In total we use 39 pairs from 10 RF, where all needed data are available for the PMM application, with a combined duration of 1786 seconds in cloud. The aerosol loading mean $N_{\mathrm{CCN}_{0.43\%}}$ values range from $96\ cm^{-3}$ in clean conditions up to $1788\ cm^{-3}$ in high polluted environments. The mean of $N_{\mathrm{C}}$ is between $208 - 1367\ cm^{-3}$. The measured $w$ distributional mean ranges from 0.25 up to $2.07\ ms^{-1}$. During RF02 15 February 2020 flight a distinct shift of $N_{\mathrm{CCN}_{0.43\%}}$ was measured between 17:42 to 17:57 UTC which can be attributed to a plume crossing event and affected pairs were excluded from the analysis, because the link between aerosol environment and measured $N_{\mathrm{C}}$ through cloud formation is questionable. The horizontal distance between aerosol measurements below cloud and cloud measurements in cloud base is mainly below $30\ km$ and never exceeds $40\ km$. Results derived from the PMM are more robust with a choice of narrow a priori boundaries for classifying similar environmental conditions.

The same procedure was applied to the flights of the ACTIVATE August 2020 deployment resulting in the pairs listed in Table 2. We use a total of 16 pairs from 5 RF with a combined duration of 360 seconds in cloud. The full data set of the ACTIVATE August-September 2020 deployment including CCN measurements is only available for the August period limiting available pairs. The reduced fraction of time in cloud is in line with the observed lower cloud fraction and horizontal dimension of clouds during summer. In addition to excluding pairs affected by precipitation the pairs in RF28 were not used in the analysis because of a smoke layer possibly altering the cloud formation process. The aerosol loading mean $N_{\mathrm{CCN}_{0.43\%}}$ values range from $122\ cm^{-3}$ in clean conditions up to $1995\ cm^{-3}$ in high polluted environments. The pairs in summer exhibit








**Table 2.** All pairs consisting of serial below cloud base (BCB) and above cloud base (ACB) legs during the August 2020 deployment. Mean values and standard deviation in parenthesis for $w$ and $N_C$ from ACB cloud portions, and $N_{CCN_{0.43\%}}$ from the BCB legs. $D_{max}$ is the maximal distance of cloud measurements to the aerosol measurements and $h_{ACB}$ is the height above cloud base with standard deviation in parenthesis.

| Flight | Date | $t_{initial}$ [UTC] | CCN-100 Mode (Supersat [%]) | $D_{max}$ [km] | in-cloud [s] | $N_{CCN_{0.43\%}}$ [$cm^{-3}$] | $w$ [$ms^{-1}$] | $N_C$ [$cm^{-3}$] | $h_{ACB}$ [m] |
|---|---|---|---|---|---|---|---|---|---|
| RF23 | 13 Aug 2020 | 14:48:15 | Scan (0.16-0.71) | 19.4 | 31 | 225 ($\pm$ 22) | 0.55 ($\pm$ 0.32) | 169 ($\pm$ 71) | 129 ($\pm$ 13) |
| RF23 | 13 Aug 2020 | 16:59:29 | Scan (0.17-0.71) | 25.1 | 55 | 267 ($\pm$ 30) | 0.39 ($\pm$ 0.27) | 145 ($\pm$ 68) | 164 ($\pm$ 2) |
| RF24 | 17 Aug 2020 | 14:54:38 | Scan (0.16-0.72) | 22.7 | 26 | 304 ($\pm$ 30) | 0.68 ($\pm$ 0.37) | 208 ($\pm$ 87) | 152 ($\pm$ 12) |
| RF24 | 17 Aug 2020 | 15:01:12 | Scan* (0.17-0.71) | 36.7 | 18 | 372 ($\pm$ 22) | 0.64 ($\pm$ 0.42) | 163 ($\pm$ 105) | 101 ($\pm$ 10) |
| RF24 | 17 Aug 2020 | 15:34:01 | Scan* (0.16-0.71) | 27.0 | 31 | 122 ($\pm$ 10) | 0.87 ($\pm$ 0.70) | 103 ($\pm$ 62) | 71 ($\pm$ 16) |
| RF24 | 17 Aug 2020 | 16:57:07 | Scan* (0.17-0.72) | 36.5 | 15 | 204 ($\pm$ 16) | 0.82 ($\pm$ 0.54) | 173 ($\pm$ 66) | 127 ($\pm$ 6) |
| RF25 | 20 Aug 2020 | 14:42:26 | Scan (0.18-0.71) | 20.0 | 18 | 1744 ($\pm$ 110) | 0.89 ($\pm$ 0.69) | 649 ($\pm$ 510) | 94 ($\pm$ 3) |
| RF25 | 20 Aug 2020 | 14:49:00 | Scan* (0.17-0.71) | 30.1 | 11 | 1586 ($\pm$ 82) | 0.95 ($\pm$ 1.16) | 658 ($\pm$ 605) | 79 ($\pm$ 2) |
| RF25 | 20 Aug 2020 | 15:13:03 | Scan* (0.17-0.71) | 35.0 | 20 | 1291 ($\pm$ 54) | 0.78 ($\pm$ 0.58) | 484 ($\pm$ 462) | 123 ($\pm$ 4) |
| RF25 | 20 Aug 2020 | 15:54:36 | Scan* (0.17-0.72) | 31.7 | 24 | 1113 ($\pm$ 76) | 0.71 ($\pm$ 0.54) | 557 ($\pm$ 449) | 89 ($\pm$ 2) |
| RF25 | 20 Aug 2020 | 16:03:21 | Scan (0.16-0.71) | 26.2 | 24 | 1266 ($\pm$ 47) | 0.72 ($\pm$ 0.48) | 739 ($\pm$ 537) | 61 ($\pm$ 3) |
| RF26 | 21 Aug 2020 | 15:35:56 | Scan (0.16-0.71) | 32.2 | 20 | 1261 ($\pm$ 78) | 0.35 ($\pm$ 0.28) | 458 ($\pm$ 373) | 57 ($\pm$ 9) |
| RF27 | 25 Aug 2020 | 15:10:29 | Scan (0.16-0.72) | 28.8 | 15 | 1627 ($\pm$ 101) | 0.84 ($\pm$ 0.66) | 338 ($\pm$ 232) | 129 ($\pm$ 4) |
| RF27 | 25 Aug 2020 | 15:19:14 | Scan (0.16-0.71) | 16.1 | 10 | 1529 ($\pm$ 91) | 0.70 ($\pm$ 0.65) | 440 ($\pm$ 324) | 136 ($\pm$ 6) |
| RF27 | 25 Aug 2020 | 16:18:18 | Scan* (0.16-0.71) | 33.4 | 19 | 1794 ($\pm$ 89) | 0.59 ($\pm$ 0.50) | 536 ($\pm$ 434) | 82 ($\pm$ 3) |
| RF27 | 25 Aug 2020 | 16:27:02 | Scan (0.16-0.72) | 35.3 | 23 | 1995 ($\pm$ 186) | 0.35 ($\pm$ 0.27) | 575 ($\pm$ 455) | 105 ($\pm$ 3) |
| All RF | Average | | | 28.5 | 23 | 1044 | 0.68 | 400 | 106 |

\* Only one scan in BCB leg.

a bimodal distribution of either very clear or high polluted conditions and are similar within a day while the high polluted conditions occur within a higher frequency. The $N_C$ mean ranges from 103 up to 739 $cm^{-3}$ which is 25% lower in terms of all pairs average of 400 $cm^{-3}$ compared to the wintertime 535$cm^{-3}$ average and in good agreement with the findings of Dadashazar et al. (2021b). A similar trend is observed in the $w$ measurements where the mean $w$ is from 0.35 to 0.95 $ms^{-1}$ and thus 33% lower in terms of all pairs average 0.68 $ms^{-1}$ in comparison with the wintertime 1.02 $ms^{-1}$ average. Also, less variability of updraft speed was measured with an average of 0.53 $ms^{-1}$(76%) compared to 0.76 $ms^{-1}$(78%) during wintertime indicating a higher dynamical influence during winter, i.e., with a high intraday variability.





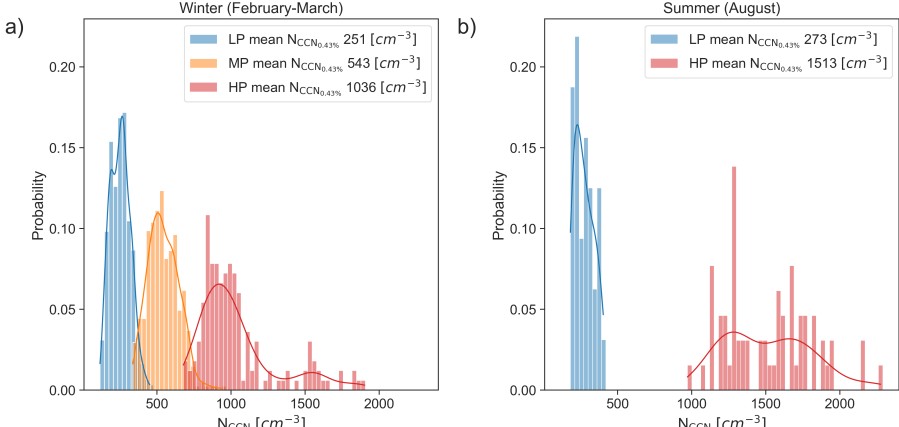

**Figure 4.** Probability distributions of wintertime $N_{CCN_{0.43\%}}$ of the low polluted (LP/blue), medium polluted (MP/orange) and high polluted (HP/red) group with their mean values (a). All values are binned with a bin width of $30\ cm^{-3}$ and the abscissa gives the probability of occurrence in the group. The same is done in b) for the summertime $N_{CCN_{0.43\%}}$ distribution of the low polluted and high polluted group. No summertime pair is attributable to the MP group and thus not shown.

## 3.2 Seasonal Aerosol Distribution and Composition below Cloud Base

The pair measurements indicate a correlation between $N_C$ and $w$ which can be quantified by the PMM. For this purpose, all pairs of a season are sorted according to their mean aerosol loading into three groups where each has a new set of $N_{CCN_{0.43\%}}$, $N_C$ and $w$ respectively. Figure 4a depicts the $N_{CCN_{0.43\%}}$ distribution of each group observed during winter. The LP group ranges from $115 - 451\ cm^{-3}$ with a mean of $251\ cm^{-3}$ for $N_{CCN_{0.43\%}}$. The MP(HP) group have values from $337(678) - 941(1903)\ cm^{-3}$ with a mean $N_{CCN_{0.43\%}}$ of $542(1036)\ cm^{-3}$ respectively. There are overlap regions between the groups due to the usage of single values of $N_{CCN_{0.43\%}}$ in Figure 4, but the separation is sufficient for the applicability of the PMM.

The same is shown for the summer period in Figure 4b, where only pairs corresponding to the LP and HP group were measured. For the summertime the LP(HP) group's $N_{CCN_{0.43\%}}$ values range from $181(971) - 403(2275)\ cm^{-3}$ with a mean of $273(1513)\ cm^{-3}$. The minimum value of the summertime HP group is noticeably higher compared to the wintertime HP group, does not have a distinct peak and is more equally distributed. The summer HP group's mean $N_{CCN_{0.43\%}}$ is 46% higher than the winter HP mean value while the LP groups are quite similar in shape. Interestingly, no pair in the MP group was measured during summer and $N_{CCN_{0.43\%}}$ is either in clean or high polluted conditions within a research flight.

$N_{CCN_{0.43\%}}$ are a subsample of the available aerosol population. The aerosol size distributions during wintertime are shown in Figure 5a. The clean marine environment (LP) has a distinct bimodal distribution consisting of an Aitken mode ($10 - 100\ nm$) peaking at around $20\ nm$ and an accumulation mode ($100 - 1000\ nm$) at $100\ nm$. In contrast the HP group has a unimodal distribution with a flat peak at $40 - 100\ nm$ at similar $dNdlogD_p$ concentrations to the LP and MP group and exhibits a plateau below $20\ nm$ which hints to an overlapping ultra fine particle mode. The integrated number concentration for particles greater than $85\ nm$ $N_{gt85}$ in the BCB leg, depicted in Figure 5c, shows that the steady increase from $472\ cm^{-3}$ (LP) over





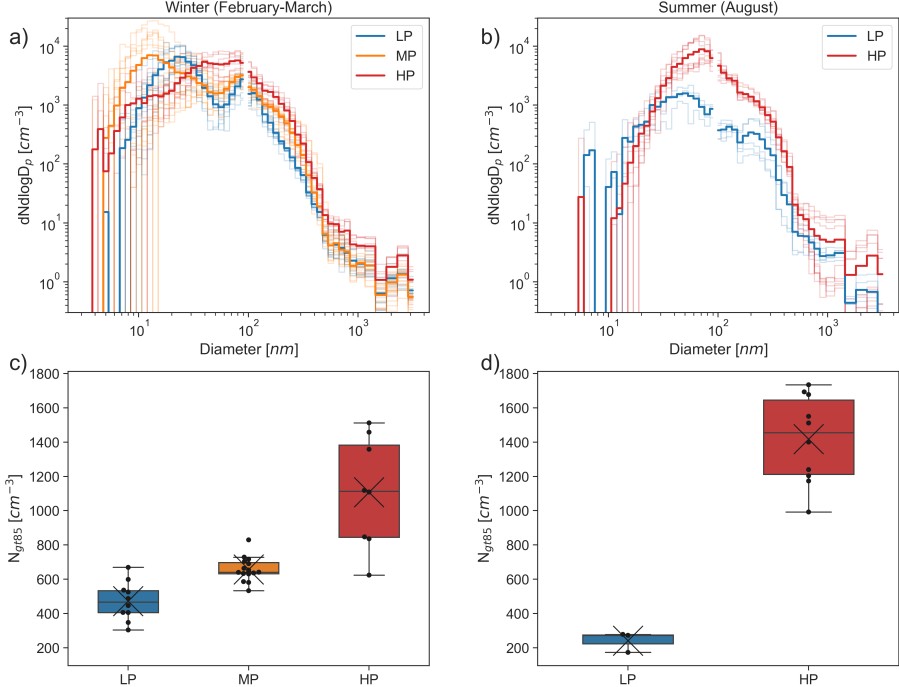

**Figure 5.** Wintertime aerosol size distributions from the SMPS/LAS instrument combination of the low polluted (LP) (blue), medium polluted (MP) (orange) and high polluted (HP) (red) group with their mean distribution and shaded distributions for a single pair (a). The same is done in b) for the summertime aerosol size distribution of the low polluted and high polluted group. Whisker plots of $N_{gt85}$ with the mean marked by a cross for wintertime (c) and summertime (d). No summertime pair is attributable to the MP group and thus not shown in b) and d).

$658\,cm^{-3}$ (MP) to $1108\,cm^{-3}$ (HP) is significant for LP to HP and probably stems from condensational growth and coagulation processes for both Aitken and accumulation modes. The unimodal HP size distribution originates from an overlapping Atiken and acumulation mode where the aerosol properties can differ. The MP group has a bimodal distribution with the maximum of the Aitken mode at around $15\,nm$ and accumulation mode peaks similarly at $100\,nm$.





**Table 3.** All pair's aerosol composition below cloud base (BCB) during the February-March 2020 deployment. Mean values and standard deviation in parenthesis for organic aerosol (OA), $SO_4^{2-}$, $NO_3^-$, $NH_4^+$ from AMS measurements and sea salt from PILS measurements. All measurements are given at standard temperature and pressure.

| Group | Flight | date | $t_{initial}$ [UTC] | SeaSalt $[\mu g m^{-3}]$ | OA $[\mu g m^{-3}]$ | $SO_4^{2-}$ $[\mu g m^{-3}]$ | $NO_3^-$ $[\mu g m^{-3}]$ | $NH_4^+$ $[\mu g m^{-3}]$ |
|---|---|---|---|---|---|---|---|---|
| LP | RF13 | 01 Mar 2020 | 14:53:22 | 1.97(±0.35) | 0.57(±0.14) | 0.93(±0.05) | 0.06(±0.04) | 0.41(±0.14) |
| LP | RF17 | 08 Mar 2020 | 14:32:31 | 2.39(±0.00) | 0.35(±0.20) | 0.30(±0.03) | 0.05(±0.02) | 0.18(±0.07) |
| LP | RF17 | 08 Mar 2020 | 14:41:44 | 4.23(±2.96) | 0.32(±0.13) | 0.35(±0.05) | 0.07(±0.04) | 0.06(±0.04) |
| LP | RF17 | 08 Mar 2020 | 15:09:24 | 3.69(±0.00) | 0.15(±0.10) | 0.36(±0.05) | 0.02(±0.03) | 0.03(±0.11) |
| LP | RF17 | 08 Mar 2020 | 15:50:27 | 4.87(±0.00) | 0.42(±0.08) | 0.39(±0.02) | 0.07(±0.03) | 0.26(±0.13) |
| LP | RF17 | 08 Mar 2020 | 15:59:48 | 4.42(±0.49) | 0.43(±0.12) | 0.42(±0.03) | 0.04(±0.05) | 0.04(±0.16) |
| LP | RF19 | 09 Mar 2020 | 17:25:14 | 3.14(±0.00) | 0.16(±0.08) | 0.30(±0.04) | 0.04(±0.03) | 0.07(±0.12) |
| LP | RF19 | 09 Mar 2020 | 17:55:32 | 3.80(±0.00) | 0.08(±0.16) | 0.34(±0.02) | 0.03(±0.04) | 0.03(±0.09) |
| LP | RF19 | 09 Mar 2020 | 18:39:22 | 3.80(±0.13) | 0.27(±0.19) | 0.36(±0.03) | 0.02(±0.02) | 0.07(±0.11) |
| | LP | Average | | 3.59 | 0.31 | 0.42 | 0.04 | 0.13 |
| MP | RF01 | 14 Feb 2020 | 17:19:21 | 0.98(±0.65) | 1.20(±0.31) | 1.24(±0.05) | 2.07(±0.28) | 1.36(±0.10) |
| MP | RF01 | 14 Feb 2020 | 17:28:38 | 2.16(±0.17) | 1.26(±0.19) | 1.33(±0.16) | 1.94(±0.23) | 1.39(±0.25) |
| MP | RF01 | 14 Feb 2020 | 17:56:27 | 4.19(±0.00) | 0.92(±0.12) | 0.89(±0.06) | 0.58(±0.06) | 0.54(±0.16) |
| MP | RF01 | 14 Feb 2020 | 18:04:21 | 5.84(±0.00) | 1.08(±0.09) | 0.96(±0.07) | 0.64(±0.06) | 0.72(±0.13) |
| MP | RF02 | 15 Feb 2020 | 17:07:07 | 1.89(±0.93) | 0.80(±0.16) | 0.66(±0.06) | 0.14(±0.05) | 0.29(±0.09) |
| MP | RF02 | 15 Feb 2020 | 17:15:06 | 3.22(±0.25) | 0.99(±0.12) | 0.66(±0.05) | 0.18(±0.04) | 0.37(±0.16) |
| MP | RF02 | 15 Feb 2020 | 18:17:32 | 4.20(±1.28) | 0.83(±0.14) | 0.82(±0.05) | 0.08(±0.04) | 0.37(±0.20) |
| MP | RF02 | 15 Feb 2020 | 18:30:07 | 5.51(±0.84) | 0.64(±0.14) | 0.82(±0.05) | 0.10(±0.02) | 0.16(±0.11) |
| MP | RF09 | 27 Feb 2020 | 18:46:24 | 3.65(±0.37) | 1.56(±0.10) | 1.16(±0.07) | 0.17(±0.04) | 0.35(±0.12) |
| MP | RF09 | 27 Feb 2020 | 18:53:48 | 3.29(±0.76) | 1.41(±0.12) | 1.07(±0.02) | 0.15(±0.05) | 0.32(±0.17) |
| MP | RF09 | 27 Feb 2020 | 19:26:17 | 2.51(±0.00) | 1.44(±0.10) | 1.05(±0.07) | 0.14(±0.05) | 0.30(±0.20) |
| MP | RF09 | 27 Feb 2020 | 19:36:09 | 2.42(±0.14) | 1.40(±0.17) | 0.71(±0.06) | 0.10(±0.03) | 0.35(±0.10) |
| MP | RF09 | 27 Feb 2020 | 20:08:09 | 2.03(±0.00) | 1.70(±0.07) | 1.04(±0.05) | 0.13(±0.04) | 0.35(±0.15) |
| MP | RF09 | 27 Feb 2020 | 20:16:39 | 2.80(±0.00) | 1.54(±0.09) | 1.01(±0.04) | 0.10(±0.02) | 0.48(±0.15) |
| MP | RF13 | 01 Mar 2020 | 16:00:14 | 1.48(±0.12) | 0.81(±0.12) | 0.73(±0.04) | 0.60(±0.05) | 0.58(±0.12) |
| MP | RF21 | 12 Mar 2020 | 14:41:07 | 1.94(±0.00) | 2.09(±0.21) | 0.81(±0.04) | 0.14(±0.05) | 0.37(±0.12) |
| MP | RF21 | 12 Mar 2020 | 14:48:49 | 2.39(±0.00) | 1.79(±0.23) | 0.87(±0.05) | 0.16(±0.04) | 0.36(±0.13) |
| MP | RF21 | 12 Mar 2020 | 15:17:07 | 1.90(±0.00) | 1.66(±0.26) | 0.68(±0.04) | 0.13(±0.03) | 0.25(±0.10) |
| MP | RF21 | 12 Mar 2020 | 16:03:43 | 1.00(±0.03) | 1.75(±0.11) | 0.78(±0.05) | 0.12(±0.04) | 0.31(±0.06) |
| MP | RF21 | 12 Mar 2020 | 16:11:45 | 0.99(±0.04) | 1.73(±0.17) | 0.84(±0.04) | 0.10(±0.02) | 0.40(±0.05) |
| | MP | Average | | 2.72 | 1.33 | 0.91 | 0.39 | 0.48 |


| Group | Flight | date | $t_{initial}$ [UTC] | SeaSalt $[\mu g m^{-3}]$ | OA $[\mu g m^{-3}]$ | $SO_4^{2-}$ $[\mu g m^{-3}]$ | $NO_3^-$ $[\mu g m^{-3}]$ | $NH_4^+$ $[\mu g m^{-3}]$ |
|---|---|---|---|---|---|---|---|---|
| HP | RF03 | 17 Feb 2020 | 17:39:28 | 1.76(±0.59) | 2.94(±0.18) | 1.04(±0.10) | 0.49(±0.04) | 0.66(±0.18) |
| HP | RF13 | 01 Mar 2020 | 14:08:38 | 1.27(±0.00) | 0.99(±0.21) | 0.66(±0.09) | 0.96(±0.19) | 0.62(±0.17) |
| HP | RF16 | 06 Mar 2020 | 19:32:13 | 3.65(±0.00) | 0.93(±0.16) | 0.64(±0.06) | 0.05(±0.03) | 0.07(±0.06) |
| HP | RF16 | 06 Mar 2020 | 19:40:24 | 6.72(±0.00) | 2.19(±0.18) | 0.91(±0.04) | 0.23(±0.04) | 0.45(±0.08) |
| HP | RF16 | 06 Mar 2020 | 20:13:19 | 4.95(±1.45) | 1.93(±0.16) | 0.96(±0.07) | 0.13(±0.04) | 0.44(±0.13) |
| HP | RF16 | 06 Mar 2020 | 20:22:54 | 7.26(±0.00) | 0.81(±0.10) | 0.56(±0.05) | 0.03(±0.04) | 0.20(±0.08) |
| HP | RF20 | 11 Mar 2020 | 13:44:36 | 3.25(±0.25) | 1.25(±0.12) | 0.29(±0.02) | 0.18(±0.03) | 0.07(±0.11) |
| HP | RF20 | 11 Mar 2020 | 14:23:48 | 3.70(±0.00) | 1.70(±0.15) | 0.38(±0.03) | 0.18(±0.05) | 0.08(±0.08) |
| HP | Average | | | 4.07 | 1.59 | 0.68 | 0.28 | 0.32 |

Since the groups are categorized by their mean $N_{CCN_{0.43\%}}$ and the group's $N_{gt85}$ are constantly higher than their $N_{CCN_{0.43\%}}$, the activation radii of the size distribution at 0.43% supersaturation is probably between $85 - 93\ nm$ for the MP and HP group and around $106\ nm$ for the LP group. The winter groups differ for particles smaller than $40\ nm$, which contributes a high fraction to the available aerosol population for the LP and MP group. We consider particles smaller than $40\ nm$ as irrelevant for the cloud formation process itself, but as a critical reservoir for the accumulation mode through altering processes, which can be seen in the HP group's distribution. The MP group with its high fraction of particles below $20\ nm$ could hint to the process of new particle formation (Zheng et al., 2021). However, the aerosol size distributions display that for a critical activation radii down to $40\ nm$ the HP group has the highest amount of particles being possible CCN, followed by the MP group and finally the LP group.

During summertime the aerosol size distribution of the LP and HP group are comparable by adhering to a unimodal distribution, but differ significantly in the $dNdlogD_p$ concentrations between $10\ nm$ and $400\ nm$ (see Figure 5b). This difference is reflected in $N_{gt85}$ in Figure 5d with mean values of $241\ cm^{-3}$ (LP) compared to $1418\ cm^{-3}$ (HP). The summer group's mean $N_{gt85}$ is smaller than their mean $N_{CCN_{0.43\%}}$, suggesting a critical activation radius of the size distribution below $85\ nm$ at 0.43% supersaturation for both groups. Here the altering processes of Aitken and accumulation modes are negligible and the difference in the HP group suggest another source of pollution during summer. The WNAO is directly located in the Northern hemisphere west wind band in winter, but during summertime the anticyclonic circulation driven by the Bermuda-Azores High influences the study region with a south west wind component (Sorooshian et al., 2020; Painemal et al., 2021; Dadashazar et al., 2021a). Therefore the sources of pollution can change between the seasons.

The wintertime aerosol mass concentrations in the BCB legs are given in Table 3. Sea salt is the dominant species with respect to mass throughout the season and has a high variability day to day and within a research flight. The highest concentrations were measured during RF16 on 6 March 2020, which can be attributed to the HP group and thus yield high $N_{CCN_{0.43\%}}$. However, there is no observable trend of sea salt mass concentration between the groups. On the other hand OA shows a significant increase from the LP to the MP/HP group. It can be deduced that the MP and HP group are influenced by pollution sources like





**Table 4.** All pair's aerosol composition below cloud base (BCB) during the August 2020 deployment. Mean values and standard deviation in parenthesis for organic aerosol (OA), $SO_4^{2-}$, $NO_3^-$, $NH_4^+$ from AMS measurements and sea salt from PILS measurements. All measurements are given at standard temperature and pressure.

| Group | Flight | date | $t_{initial}$ [UTC] | SeaSalt $[\mu g m^{-3}]$ | OA $[\mu g m^{-3}]$ | $SO_4^{2-}$ $[\mu g m^{-3}]$ | $NO_3^-$ $[\mu g m^{-3}]$ | $NH_4^+$ $[\mu g m^{-3}]$ |
|---|---|---|---|---|---|---|---|---|
| LP | RF23 | 13 Aug 2020 | 14:43:44 | - | 0.25(±1.00) | 1.32(±0.11) | 0.04(±0.11) | 0.26(±0.21) |
| LP | RF23 | 13 Aug 2020 | 16:56:11 | - | 0.78(±0.62) | 1.19(±0.06) | 0.07(±0.04) | -0.13(±0.63) |
| LP | RF24 | 17 Aug 2020 | 14:51:54 | 1.62(±0.00) | 2.18(±0.25) | 1.31(±0.14) | 0.16(±0.06) | 0.48(±0.37) |
| LP | RF24 | 17 Aug 2020 | 15:00:00 | 4.32(±0.00) | 1.81(±0.37) | 1.39(±0.14) | 0.14(±0.09) | -0.12(±0.52) |
| LP | RF24 | 17 Aug 2020 | 16:55:16 | 0.24(±0.00) | 1.22(±0.21) | 0.67(±0.08) | 0.08(±0.04) | -0.27(±0.28) |
| | LP | Average | | 2.06 | 1.25 | 1.17 | 0.10 | 0.04 |
| HP | RF25 | 20 Aug 2020 | 14:40:06 | 0.75(±0.20) | 10.38(±0.28) | 3.24(±0.15) | 0.64(±0.12) | 1.29(±0.43) |
| HP | RF25 | 20 Aug 2020 | 14:47:13 | 0.44(±0.24) | 9.88(±0.48) | 3.50(±0.13) | 0.40(±0.07) | 1.31(±0.33) |
| HP | RF25 | 20 Aug 2020 | 15:12:04 | 0.60(±0.00) | 7.40(±0.29) | 2.53(±0.08) | 0.26(±0.08) | 0.78(±0.31) |
| HP | RF25 | 20 Aug 2020 | 15:53:07 | 1.71(±0.00) | 6.18(±0.52) | 2.28(±0.10) | 0.23(±0.10) | 1.08(±0.22) |
| HP | RF25 | 20 Aug 2020 | 16:01:02 | 6.25(±0.00) | 7.10(±0.43) | 2.55(±0.07) | 0.24(±0.05) | 1.05(±0.21) |
| HP | RF26 | 21 Aug 2020 | 15:32:54 | 0.41(±0.00) | 5.33(±0.55) | 2.45(±0.08) | 0.22(±0.07) | 1.05(±0.33) |
| HP | RF27 | 25 Aug 2020 | 15:07:52 | 3.53(±0.00) | 8.91(±0.55) | 1.84(±0.08) | 0.47(±0.07) | 0.77(±0.48) |
| HP | RF27 | 25 Aug 2020 | 15:16:54 | 4.25(±0.00) | 9.03(±0.32) | 2.10(±0.11) | 0.44(±0.09) | 0.40(±0.31) |
| HP | RF27 | 25 Aug 2020 | 16:16:47 | 4.04(±0.88) | 9.66(±0.41) | 1.66(±0.08) | 0.48(±0.05) | 0.40(±0.32) |
| HP | RF27 | 25 Aug 2020 | 16:24:52 | 3.21(±0.06) | 9.67(±0.58) | 1.71(±0.09) | 0.47(±0.11) | 0.72(±0.21) |
| | HP | Average | | 2.52 | 8.35 | 2.39 | 0.39 | 0.89 |

the North East Coast American outflow, while the LP group represents natural marine conditions. The $SO_4^{2-}$, $NO_3^-$ and $NH_4^+$

325    mass concentrations have a slight increase from the clean marine condition (LP) to polluted conditions (MP/HP) and i.e. RF01 on 14 February 2020 is an outlier and has the highest values, which decreases farther offshore during the flight.

In Table 4 is the BCB aerosol mass concentration below cloud depicted for the August 2020 summertime period. The sea salt mass concentration is highly variable like wintertime with low statistics in the LP group. Negative values for $NH_4^+$ mean that the mass concentration is lower than the calibrated background concentration and thus real. A significant increase from the LP

330    to the HP group is measured for all species expect sea salt and suggest more pollution in the summer season. The difference in mass concentration is not equally distributed with the smallest rate of a doubling for $SO_4^{2-}$, followed by a factor of 4 for $NO_3^-$ and a factor of over 6(20) for OA($NH_4^+$). The chemical composition of the aerosol population alters $N_C$ (Hoose and Möhler, 2012), i.e. the organic carbon species have variable influences depending on solubility, molecular weight and surface tension (Ervens et al., 2005).





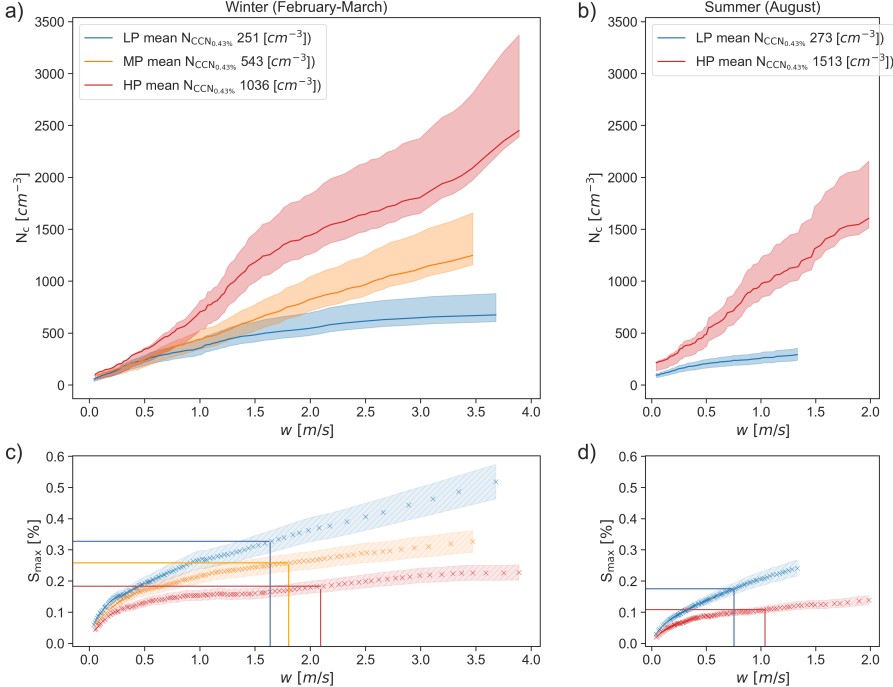

**Figure 6.** a) The lines represent the $w$ to $N_C$ relation derived with the PMM for the low polluted (LP/blue), medium polluted (MP/orange) and high polluted (HP/red) group with their boundaries of mean $N_{CCN_{0.43\%}}$ in parenthesis. The dark shaded areas represent the measurement uncertainty of 20% in addition with the relative error calculated according to Haddad and Rosenfeld (1997) with the assumption that the standard deviation of $N_C$ in each group represents the ratio of the noisy variation in the $N_C$ measurements to the true variation in $N_C$. c) The $S_{max}$ estimate of each group is given in 'x' markers for the same $w$ spectrum with the error as shallow lined shaded area. The vertical lines are the $w_{eff}$ with associated $S_{max}$. The same PMM and $S_{max}$ analysis in b) and d) for the summertime LP and HP group.

### 3.3 Seasonal impact of $w$ and $N_{CCN_{0.43\%}}$ on $N_C$

Figure 6a shows the application of the PMM to all groups of the winter season. The $w$ to $N_C$ relations shows the fraction of activated aerosol from the aerosol size distribution for a given updraft of supersaturation, respectively. The LP group, which has a mean $N_C$ of $315(\pm 165)\ cm^{-3}$, shows the highest impact of $w$ to $N_C$ for $w < 1.4\ ms^{-1}$ and reaches saturation for higher $w$ values. The MP group exhibits a similar trend with a mean of $518(\pm 304)\ cm^{-3}$, but the impact of $w$ is decreasing slower compared to the LP group for higher $w$. The HP group shows the strongest impact for $w < 1.6\ ms^{-1}$ and as a mean $N_C$ of $930(\pm 630)\ cm^{-3}$. In addition, it has a second mode with a strong increase in $N_C$ for $w > 3\ ms^{-1}$.

The two domains of $w$ in the HP group could represent the activation of smaller aerosol particles from the aerosol population. Since the critical diameter of aerosol activation depends on the supersaturation and is shifted towards smaller diameters for higher supersaturation, the positive correlation of $w$ and supersaturation results in smaller aerosols getting activated for higher $w$ (Köhler, 1936; Dusek et al., 2006; Schulze et al., 2020). $N_C$ are slightly smaller than the respective group's $N_{CCN_{0.43\%}}$



leading to a mean supersaturation below 0.43% in winter. The LP group exhibits some characteristics of an aerosol-limited regime with $N_C$ highly depending on the available aerosol population, while the HP group shows the characteristics of an updraft-limited regime with $N_C$ being directly proportional to $w$ (Reutter et al., 2009). The MP group is between both regimes and tend to the characteristics of an updraft-limited regime, since $N_C$ does not reach saturation for high $w$.

350    The $S_{max}$ estimate for each group's $w_{eff}$ in Figure 6c is decreasing with increasing pollution level and are 0.33%(LP), 0.26%(MP) and 0.18%(HP), respectively. Since the variability of updraft speed is higher with larger $w$, the local supersaturation can deviate from the derived $S_{max}$ estimates. The reduction of $S_{max}$ for increasing pollution levels demonstrate the water vapor competition of more activated CCN and thus function as a buffer for preventing higher supersaturation. The LP group's mean $N_C$ is above its mean $N_{CCN_{0.43\%}}$ although $S_{max}$ is near, and below 0.43%, which could be explained by a contribution of the soluble Aitken mode particles in the bimodal aerosol size distribution (Pöhlker et al., 2021). However, the winter groups exhibit mean $N_C$ near $N_{CCN_{0.43\%}}$ with a trend of a reduced fraction of activated aerosol with increasing pollution level.

In Figure 6b the PMM is applied to the summer season in the same way. The impact of $w$ on $N_C$ has a similar trend in summer and winter for the LP group up to the maximal measured $w$ of $1.3 \ ms^{-1}$ during summer and has a mean $N_C$ of $196(\pm55) \ cm^{-3}$. The HP group has a nearly constant impact for the full range of $w$ up to $2.1 \ ms^{-1}$ and a mean $N_C$ of 360    $642(\pm389) \ cm^{-3}$. The $w$ to $N_C$ relation coincides with the wintertime equivalent for $w$ below $1.7 \ ms^{-1}$. The $S_{max}$ estimate for each group's $w_{eff}$ in Figure 6d is analogously reduced from the LP to the HP group in summer as in winter, while between the seasons a halfing of the $S_{max}$ takes place.

$N_{gt85}$ of the summer LP group is significantly lower than its winter counterpart, thus less aerosol for cloud formation is available in clean conditions during summer compared to winter. On the other hand $N_{gt85}$ of the HP summer group is 365    substantially higher than during winter. Another key feature is the lower mean $N_{gt85}$ in comparison to the mean $N_{CCN_{0.43\%}}$, showing a higher fraction of activated CCN in summer for a given supersaturation of 0.43%, which hints to a lower mean critical supersaturation needed for activation of the summer aerosol composition. Table 3 and Table 4 show an increased mass concentration of OA and $SO_4^{2-}$ between the respective groups. The high hygroscopicity of $SO_4^{2-}$ is most likely accountable for the observed lower mean $N_{gt85}$ than mean $N_{CCN_{0.43\%}}$, because the raised OA mass concentrations from the LP to HP group is 370    not reflected. Lower supersaturation in summer due to the smaller updrafts results in less activated CCN. The bisection of $w_{eff}$ in Figure 6b propagates through derived $S_{max}$ estimates to $N_C$.

## 4    Conclusions

In this study we examine the seasonal impact of $w$ and $N_{CCN_{0.43\%}}$ on $N_C$ over the WNAO from an in-situ perspective during the ACTIVATE campaign. The impact is determined by a statistical approach with the PMM where pairs of flight legs below 375    and in cloud base are used to categorize in-situ measurements into similar environmental conditions and $N_{CCN_{0.43\%}}$. We also give detailed information on the aerosol size distribution and composition below cloud base. Key findings are summarized and related to 2020 winter (February-March) and summer (August) conditions as follows:





- N$_C$ in low clouds over the WNAO show a positive correlation with $w$ and N$_{CCN_{0.43\%}}$. Updrafts smaller than $1.3\ ms^{-1}$ have the highest impact on N$_C$ in both seasons. Polluted environments exhibit a stronger $w$ impact over the full $w$ distribution in a season, while in clean marine environments the available N$_{CCN}$ limit N$_C$ for higher $w$.

- The WNAO exhibits an anti-correlated seasonal cycle of N$_C$ and N$_{CCN_{0.43\%}}$ at cloud base with 25% less N$_C$ and 71% more N$_{CCN_{0.43\%}}$ in their overall observed mean values in summer compared to winter. The seasonal cycle is consistent with the anti-correlated AOD and N$_C$ cycle measured by remote sensing and satellite instruments (Dadashazar et al., 2021b).

- The mean values of $w$ at cloud bases are 33% lower in summer compared to winter. Simultaneously the variability of updraft speeds is reduced by 31% in summer. Both indicate a higher dynamical influence during winter. A correlation of N$_C$ and $w$ is observed in the seasonal cycle and suggest that the difference between the seasons is driven by dynamics.

- The winter N$_{CCN_{0.43\%}}$ directly below cloud shows a broad distribution due to different aerosol sources and pollution levels, while only clear sky or high polluted conditions were measured in summer. For high polluted environments, summer exhibits a 46% increased mean N$_{CCN_{0.43\%}}$.

- The aerosol size distribution during winter exhibits a bimodal distribution in clean marine and medium polluted condition, which transforms into a unimodal distribution for higher pollution levels. The Aitken mode acts as reservoir for the accumulation mode, since N$_{gt85}$ increases while the aerosol number concentrations do not differ significantly. In contrast to the winter period, the summer period is characterized by unimodal distributions and a clear difference between the aerosol concentrations of the pollution levels.

- The aerosol composition shows a constant proportion of sea salt in each season, with an increased aerosol mass concentration measured in winter, which could be related to the increased surface wind speeds resulting in more efficient wind-driven sea salt emissions (Painemal et al., 2021). With the increase in pollution levels, a concomitant increase in OA, $SO_4^{2-}$, $NO_3^-$ and $NH_4^+$ mass concentrations is measured in summer. In winter, the increase is comparatively moderate.

- $w$ and related S$_{max}$ determine the range of activated CCN and S$_{max}$ is reduced at increasing pollution levels. As shown, $w$ dominantly affects the activation of CCN and determines the fraction of activated aerosol and thus explains generally higher N$_C$ values during winter compared to summer.

The observational data presented in this study includes key parameters which are used in state-of-the-art aerosol-climate models to describe aerosol-induced cloud modifications. Consistent observations of the aerosol number concentration, size distribution and composition, $w$ as well as N$_C$ are provided for a wide range of conditions in the winter and summer seasons. Hence the data could serve as a valuable basis for evaluating and further improving the representation of aerosol-cloud interactions in future climate simulations.



*Data availability.* The ACTIVATE data are available at http://doi.org/10.5067/SUBORBITAL/ACTIVATE/DATA001

# Appendix A


## A1 List of symbols and abbreviations

| | |
|---|---|
| $D_{max}$ | maxmimum distance of cloud measurements to aerosol measurements |
| $h_{ACB}$ | height above cloud base |
| $N_C$ | cloud droplet number concentration |
| $N_{CCN_{0.43\%}}$ | cloud condensation nuclei concentration at 0.43% supersaturation |
| $S_{max}$ | maximum supersaturation in cloud base |
| $w$ | updraft speed |
| $w_{eff}$ | effective updraft speed |
| 2D-S | Two-Dimensional Stereo probe |
| ACB | above cloud base |
| ACTIVATE | Aerosol Cloud meTerology Interactions oVer the western ATlantic Experiment |
| AMS | Aerosol Mass Spectrometer |
| AOD | aerosol optical depth |
| BCB | below cloud base |
| CCN | cloud concensation nuclei |
| CMIP6 | Coupled Model Intercomparison Project Phase 6 |
| DoF | depth of field |
| FCDP | Fast Cloud Droplet Probe |
| LAS | Laser Aerosol Spectrometer |
| MBL | marine boundary layer |
| OA | organic aerosol |
| PAS | particle air speed |
| PILS | Particle-Into-Liquid Sampler |
| PMM | Probability Matching Method |
| RF | research flights |
| SMPS | Scanning Mobility Particle Sizer |
| SPEC Inc. | Stratton Park Engineering Company Incorporated |
| TAMMS | turbulent air motion measurement system |
| WNAO | Western North Atlantic Ocean |





*Author contributions.* S.K. conducted the analysis and wrote the manuscript. C.V.advised the study and provided intensive feedback on the manuscript. K.T. wrote section 2.2.3. A.S. and L.Z. wrote section 2.2.5. L.Z. wrote section 2.2.6. J.H. contributed to section 1 and 4. E.C, E.W and L.Z. performed the flight measurements. S.K., V.H., St.K., C.R. and D.S. participated in instrument calibration. B.A., R.M., L.Z. and A.S. participated in mission planning. R.F., A.J.S and M.S. conducted the weather forecast. E.C, R.F., A.J.S., R.M., M.S. and A.S. participated in strategic flight planning. S.K., R.M, K.T., C.R, E.W., L.Z., M.S. and A.S. participated in mission operation. G.C. and M.S. conducted the data management. All authors commented on the manuscript.

*Competing interests.* The authors declare that they have no conflict of interest.

*Acknowledgements.* The work was funded by NASA grant 80NSSC19K0442 in support of ACTIVATE, a NASA Earth Venture Suborbital-3 (EVS-3) investigation funded by NASA's Earth Science Division and managed through the Earth System Science Pathfinder Program Office. C.V, S.K. and St.K. were funded by the Helmholtz excellence programme (grant number W2/W3-060, by the Deutsche Forschungsgemein­schaft (DFG, German Research Foundation) – TRR 301 – Project-ID 428312742 and the SPP 1294 HALO under contract VO 1504/7-1.





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
