# Peer review of "Seasonal updraft speeds change cloud droplet number concentrations in low level clouds over the Western North Atlantic"

_Atmospheric Chemistry and Physics, 2022_

## Referee Comment (RC1)

The research describes in-situ measurements and analysis from the ACTIVATE project over the Western North Atlantic Ocean. Investigation of low level cloud layers and measurements of Cloud Condensation Nuclei (CCN) concentrations below cloud base were compared with droplet number concentrations (CNDC) from just above cloud base. They present this in the context of the vertical wind speed (w) and use a method to relate the updraft velocity to the CDNC just above cloud base.

They found a significant range of CCN concentrations that spanned different seasons. One of the key aspects of the paper split the measured CCN into Low Polluted (LP), Medium Polluted (MP) and High Polluted (HP) groups. The authors found all occurred in the winter while summer was found to be split between LP and HP.

The contributions to the CCN population from the aerosol composition was presented, with the conclusion that Sea Salt Aerosol, although present in significant quantities throughout, was not the driver of the changes in aerosol groups, but rather always present as a background of deliquesced aerosol likely based on the boundary layer windspeeds. During elevated aerosol periods the strongest association was with Organic Aerosol (OA).

They found that the relationship between CCN and CDNC was due to the interplay between the dynamics and availability of CCN for the formation of cloud droplets. For example in the HP cases CDNC was sensitive to the full range of w, where as in more pristine LP cases the CDNC was CCN limited and therefore didn't change as significantly with increasing w values.

I found the paper to be well presented with excellent figures that highlighted the key aspects of the study. I recommend publication with minor corrections -

**Minor comments**

Section title and numbering missing immediately after the abstract. (should be 1. Introduction?)

**L99** *"The FCDP with its fast 100 electronics, small pinhole feature for coincidence reduction and applicable filtering techniques can be classified among the lower end of both propagated uncertainties in size and NC. "*

Has this been demonstrated or is it just an estimation?

**L104** It's stated the effective pixel size for the 2D-S is 11.4 $\mu$m. Lawson et al. (2006) describes this differently. The effective pixel width used as standard is 10 $\mu$m. Please confirm whether your effective size range is different or whether this needs correcting.

**L211** I'm not suggesting this should be changed but I've always found the terminology of 'polluted' to be a little misleading. What exactly *is* polluted? Or is it just the same as 'elevated' aerosol.

**Figure 4.** What type of fit are the lines to the histograms?

**Figure 6.** a) should maybe refer to a-b)

**General Comment:** it would be nice to see some FCDP, 2DS size distributions comparisons. The image plot 3b is fine though otherwise.

**APPENDIX:** $N_{gt85}$ definition missing.

---

## Author Response (AR1)

**Answers to reviewers**

**Reviewer #1:**

The research describes in-situ measurements and analysis from the ACTIVATE project over the Western North Atlantic Ocean. Investigation of low level cloud layers and measurements of Cloud Condensation Nuclei (CCN) concentrations below cloud base were compared with droplet number concentrations (CNDC) from just above cloud base. They present this in the context of the vertical wind speed (w) and use a method to relate the updraft velocity to the CDNC just above cloud base. They found a significant range of CCN concentrations that spanned different seasons. One of the key aspects of the paper split the measured CCN into Low Polluted (LP), Medium Polluted (MP) and High Polluted (HP) groups. The authors found all occurred in the winter while summer was found to be split between LP and HP.

The contributions to the CCN population from the aerosol composition was presented, with the conclusion that Sea Salt Aerosol, although present in significant quantities throughout, was not the driver of the changes in aerosol groups, but rather always present as a background of deliquesced aerosol likely based on the boundary layer windspeeds. During elevated aerosol periods the strongest association was with Organic Aerosol (OA).

They found that the relationship between CCN and CDNC was due to the interplay between the dynamics and availability of CCN for the formation of cloud droplets. For example in the HP cases CDNC was sensitive to the full range of w, where as in more pristine LP cases the CDNC was CCN limited and therefore didn't change as significantly with increasing w values.

I found the paper to be well presented with excellent figures that highlighted the key aspects of the study. I recommend publication with minor corrections –

We thank the reviewer for the insightful summary and the positive evaluation of the manuscript.

**Minor comments**

Section title and numbering missing immediately after the abstract. (should be 1. Introduction?)

We thank the reviewer for catching the missing first part of the Introduction. It seems that one page of the first part of the Introduction with section title and numbering is missing in the preprint version.

**L99** *"The FCDP with its fast 100 electronics, small pinhole feature for coincidence reduction and applicable filtering techniques can be classified among the lower end of both propagated uncertainties in size and NC. "*
Has this been demonstrated or is it just an estimation?

We thank the reviewer for finding this ambiguity and have changed the text as follows: *"The FCDP with its fast electronics, small pinhole feature for coincidence reduction and applicable filtering techniques is estimated to be among the lower end of both propagated uncertainties in size and $N_C$. "*

**L104** It's stated the effective pixel size for the 2D-S is 11.4 μm. Lawson et al. (2006) describes this differently. The effective pixel width used as standard is 10 μm. Please confirm whether your effective size range is different or whether this needs correcting.

The effective pixel size of the 2D-S was calibrated with a spinning disc experiment in the laboratory before the ACTIVATE deployment. For clarification we have changed the text to: *"It measures single particles in a size range of 5.7 - 1465 μm with a calibrated effective pixel size of 11.4 μm for each photodiode channel."*

**L211** I'm not suggesting this should be changed but I've always found the terminology of 'polluted' to be a little misleading. What exactly *is* polluted? Or is it just the same as 'elevated' aerosol.

We use 'polluted' for regions with high aerosol loading and in particular for cloud condensation nuclei concentrations above 372 /ccm derived from the statistical distribution of the measurement conditions in contrast to clean conditions with lower CCN concentrations. We agree on the misleading terminology and changed the wording of the different CCN concentrations into low (L), medium (M) and high (H) CCN concentrations at 0.43% supersaturation.

**Figure 4.** What type of fit are the lines to the histograms?

We thank the reviewer for addressing the missing fit method. We use the seaborn python package kernel density estimation for this plot which utilizes a gaussian kernel without discrete bins but in relation to the underlying 30-micron binned histogram. We added the reference:

Waskom, M. L., (2021). seaborn: statistical data visualization. Journal of Open Source Software, 6(60), 3021, https://doi.org/10.21105/joss.03021,

and added the following text to Figure 4 caption: *"The line fit represents a kernel density estimation of the python seaborn package (Waskom, 2021)."*

**Figure 6.** a) should maybe refer to a-b)

We thank the reviewer for this suggestion and have changed the Figure 6 caption to a-b) and c-d). In addition, we have deleted the last sentence of the caption as it became unnecessary.

**General Comment:** it would be nice to see some FCDP, 2DS size distributions comparisons.
The image plot 3b is fine though otherwise.

We than the reviewer for this thoughtful comment and understand that this would be viable information. Therefore, we added the mean particle size distribution of only in cloud seconds of the ACB leg for the FCDP, 2D-S and the combination of both in a separate plot below and changed the last sentence of the caption into: *"b) Histogram showing the color-coded log-normalized number*

*concentrations per bin on a 1-second basis of the 2D-S/FCDP combination with the diameter given in the ordinate and the derived mean particle size distribution of in cloud seconds during the ACB leg below."*

**APPENDIX:** $N_{gt85}$ definition missing.

We have added the $N_{gt85}$ definition to the A1 List of symbols and abbreviations.

**Reviewer #2:**

This manuscript investigates the influences of vertical updraft speeds and CCN concentration of cloud droplet number concentrations using air craft measurements. The conducted measurements and analytical methods (while not fully explained, see my comment below) appear scientifically sound. The paper is not particularly original, but can still be considered as a useful contribution to the scientific community. There are a few issues that need to be addressed better before I can recommend accepting this paper for publication.

We thank the reviewer to assess our manuscript as a useful contribution to the scientific community.

**Major comments**

The authors define three pollution levels, or regimes, based on measured CCN concentrations. There are a few issues related to this approach requiring better justifications, or some revisions. First, although it makes sense to use CCN as a measure of pollutions for the purposes of this paper, this wording is problematic considering that for air quality community pollution levels are usually defined based on concentrations of a selected chemical compounds, or simply PM mass concentration. This causes also confusing statements, such as that on lines 398-400: for air quality people it sound strange to claim that the mass concentrations of the most typical particulate air pollutants increased only moderately as the pollution levels increased in winter. I would encourage the authors to reconsider what to call the different CCN regimes used in this paper. Second, what is bases for the grouping of the different CCN regimes? The borders between the different groups given on lines 214-215 seem rather arbitrary to me. Third, if strict limits for different CCN groups are given (lines 215-215) how is it then possible that CCN concentration in different groups can overlap each other (Figure 4a and text on lines 278-380)? Finally, do the authors have any idea on why the medium CCN regime was absent during the winter?

We thank the reviewer for the detailed comment and changed the wording of the different CCN concentrations into low (L), medium (M) and high (H) CCN concentrations at 0.43% supersaturation instead of using pollution levels.

The procedure of defining the borders is given in line 211 and is based on a histogram of all pairs mean NCCN0.43% values. We changed the text to: 'For comparison both seasons share the boundaries separating the groups and the bin boundaries are chosen by identifying minima between the modes in the distribution of all winter pair mean NCCN0.43% values.', to emphasize that the borders are related to the minima between the modes.

Figure 4 does not show the group's mean NCCN0.43% distributions, but each group's set of all NCCN0.43% measurement points. We have chosen to show the single measurement points instead of the mean, because it contains more information on the underlying atmospheric data. Mean values derived from a broader distribution lead to a larger overlap region in Figure 4. So Figure 4 shows the individual datapoints and not the mean values. The quality of the separation of the groups can be assessed by the information of the overlap regions.

The absence of the medium CCN regime in summer is a very good question. The clear distinction of low and high regime in the aerosol size distribution and composition during summer shows two clearly separated aerosol sources. In contrast, in winter the presence of the medium regime hints to additional aerosol processes or a different source. The origin of the CCN sources is still unclear and is a subject of investigation in a future study. We want to note that the statistics in summer is lower compared to the winter season and we cannot exclude that the medium regime was not sampled during the six summer flights investigated here.

The description of the methodology used to determine w and Smax needs to be expanded (lines 218-223). How is Smax determined in practice? It is unclear whether w or weff is really used later in the paper, as weff appear only in equation 2. If this methodology has been described in more detail in earlier literature, the relevant studies should, at the very least, cited here properly.

We thank the reviewer for this helpful comment. We shifted the definition and reference to Smax previously given in the section 2.2.4. to the methodology section where it is most appropriate and described the procedure in more detail. We added the sentence: "We determine all groups C value with their mean cloud base temperature and pressure, a coefficient of air heat conductivity of 26.2 mW K-1 m-1 and a coefficient of water vapor diffusion in the air of 0.219 cm2 s-1 at 0° C in winter and 0.242 cm2 s-1 at 20° C in summer, see detailed mathematical background in Pinsky et al. (2012). With the help of the w to NC relation from the PMM we derive a corresponding relation of the Smax estimate (Braga et al., 2017a).".

For all updraft speed values in the Tables and in the PMM the whole w distribution of either flight segments or the group is used, if not stated otherwise. The weff is solely used to approximate a group's representative updraft speed with which the corresponding Smax estimate in Figure 6c and 6d is derived.

I have a hard time to understand how the values of Smax given on lines 350-351 have been obtained, and how they are related to the lines in figure 6c. This problem is, at least partly, related to the lack description how Smax has been determined in this study.

We hope that our additional explanation of Smax and its determination helped to resolve this issue in the revised version of the manuscript.

I would appreciate if the authors had listed concrete scientific goals for this paper. Written like it is now (lines 59-61), a reader might get an impression that the purpose of this paper is solely to produce data for other researchers for e.g model evaluation.

The goal of our paper is to show the influence of updraft speed and CCN concentrations on number concentrations of low level clouds as clearly given in the abstract. Also, in the summary we clearly state our results and differentiate the impact of updraft speeds and CCN concentrations in the different seasons. In addition, in the introduction address the relevance of our results for modeling aerosol cloud interactions.

**Minor/technical comments**

line 255: … between 208 and 1367 …

We have changed the line accordingly to the reviewer's suggestion.

line 293: please separate Ngt85 from the rest of the text using commas (i.e. …, Ngt85, …).

We separated Ngt85 as suggested.

lines 305 and 315: the term altering processes sounds odd to me. I suppose the authors mean a combination of chemical and physical (aerosol) processes that convert sub-CCN size (nucleation and Aitken mode particles) into largest ones that then contribute to the CCN population.

Yes, we can confirm that the reviewer's definition was the one we wanted to express with the terminology altering processes. We changed the text to: "We consider particles smaller than 40 nm as irrelevant for the cloud formation process itself, but as a critical reservoir for the accumulation mode through chemical and physical aerosol processes that increase the particle size, which can be seen in the H group's distribution." and "Here the aerosol processes which increase the particle size of Aitken and accumulation modes are negligible and the difference in the H group suggest another aerosol source during summer.".

lines 338-341: while I agree on that the HP regime shows the strongest increases with increasing w at w < 1.6 m s-1 and at w > 3 m s-1, it is hard to see from figure 6 that there would be such border for LP or MP, or that they would clearly saturate at high values of w.

We thank the reader for this thoughtful comment and agree on the absence of a pronounced border for the winter LP group and that for the MP group saturation is not reached. We changed the text to: "The L group, which has a mean NC of 315(±165) cm-3, shows the highest impact of w to NC for smaller w values and reaches saturation for higher w values. The M group exhibits a similar trend with a mean of 518(±304) cm-3, but the impact of w is decreasing slower compared to the L group for higher w and does not reach saturation."

The units of quantities are usually written using normal text, not in italics. Furthermore, different parts of the units should be separated from each other using spaces (i.e. m s-1, not ms-1).

We thank the reviewer for the comment and changed the quantities.

The title of Section 4 should be reconsidered (Summary and Conclusions), as most of this section is simply summary of the results with only few conclusions provided.

We changed the title of Section 4 to Summary and Conclusion.